# The role of chlorine in global tropospheric chemistry

Xuan Wang[1], Daniel J. Jacob[1,2], Sebastian D. Eastham[3], Melissa P. Sulprizio[1], Lei Zhu[1], Qianjie Chen[4], Becky Alexander[5], Tomás Sherwen[6,7], Mathew J. Evans[6,7], Ben H. Lee[5], Jessica D. Haskins[5], Felipe D. Lopez-Hilfiker[8], Joel A. Thornton[5], Gregory L. Huey[9], and Hong Liao[10]

[1] School of Engineering and Applied Sciences, Harvard University, Cambridge, Massachusetts, USA
[2] Department of Earth and Planetary Sciences, Harvard University, Cambridge, Massachusetts, USA
[3] Laboratory for Aviation and the Environment, Massachusetts Institute of Technology, Cambridge, Massachusetts, USA
[4] Department of Chemistry, University of Michigan, Ann Arbor, Michigan, USA
[5] Department of Atmospheric Sciences, University of Washington, Seattle, USA
[6] Wolfson Atmospheric Chemistry Laboratories, Department of Chemistry, University of York, York, UK
[7] National Centre for Atmospheric Science, University of York, York, UK
[8] Paul Scherrer Institute, Villigen, Switzerland
[9] School of Earth and Atmospheric Science, Georgia Institute of Technology, Atlanta, GA, USA
[10] School of Environmental Science and Engineering, Nanjing University of Information Science and Technology, Nanjing, China

*Correspondence to*: Xuan Wang (wangx@seas.harvard.edu)

**Abstract.** We present a comprehensive simulation of tropospheric chlorine within the GEOS-Chem global 3-D model of oxidant-aerosol-halogen atmospheric chemistry. The simulation includes explicit accounting of chloride mobilization from sea-salt aerosol by acid displacement of HCl and by other heterogeneous processes. Additional small sources of tropospheric chlorine (combustion, organochlorines, transport from stratosphere) are also included. Reactive gas-phase chlorine Cl*, including Cl, ClO, $Cl_2$, BrCl, ICl, HOCl, $ClNO_3$, $ClNO_2$, and minor species, is produced by the HCl + OH reaction and by heterogeneous conversion of sea-salt aerosol chloride to BrCl, $ClNO_2$, $Cl_2$, and ICl. The model simulates successfully the observed mixing ratios of HCl in marine air (highest at northern mid-latitudes) and the associated $HNO_3$ decrease from acid displacement. It captures the high $ClNO_2$ mixing ratios observed in continental surface air at night, and attributes the chlorine to HCl volatilized from sea salt aerosol and transported inland following uptake by fine aerosol. The model simulates successfully the vertical profiles of HCl measured from aircraft, where enhancements in the continental boundary layer can again be largely explained by transport inland of the marine source. It does not reproduce the boundary layer $Cl_2$ mixing ratios measured in the WINTER aircraft campaign (1-5 ppt in the daytime, low at night); the model is too high at night, which could be due to uncertainty in the rate of the $ClNO_2 + Cl^-$ reaction, but we have no explanation for the high observed $Cl_2$ in daytime. The global mean tropospheric concentration of Cl atoms in the model is 620 $cm^{-3}$ and contributes 1.0% of the global oxidation of methane, 20% of ethane, 14% of propane, and 4% of methanol. Chlorine chemistry increases global mean tropospheric BrO by 85%, mainly through the HOBr + $Cl^-$ reaction, and decreases global burdens of tropospheric ozone by 7% and OH by 3% through the associated bromine radical chemistry. $ClNO_2$ chemistry drives increases in ozone of up to 8 ppb over polluted continents in winter.

## 1 Introduction

Mobilization of chloride ($Cl^-$) from sea salt aerosol (SSA) is a large source of chlorine gases to the troposphere (Graedel and Keene, 1995; Finlayson-Pitts, 2003). These gases may generate chlorine radicals with a broad range of implications for tropospheric chemistry including the budgets of ozone, OH (the main tropospheric oxidant), volatile organic compounds (VOCs), nitrogen oxides, other halogens, and mercury (Saiz-Lopez and von Glasow, 2012; Simpson et al., 2015). Only a few global models have attempted to examine the implications of tropospheric chlorine chemistry on a global scale (Singh and Kasting, 1988; Long et al., 2014; Hossaini et al., 2016) and then only with a limited representation of processes. Here we present a more comprehensive analysis of this chemistry within the framework of the GEOS-Chem chemical transport model (CTM).

Saiz-Lopez and von Glasow (2012) and Simpson et al. (2015) present recent reviews of tropospheric halogen chemistry including chlorine. Sea-salt aerosols represent a large chloride flux to the atmosphere but most of that chloride is removed rapidly by deposition. Only a small fraction is mobilized to the gas phase as HCl or other species. Additional minor sources of tropospheric chlorine include open fires, coal combustion, waste incineration, industry, road salt application, fugitive dust, and ocean emission of organochlorine compounds (Lobert et al., 1999; McCulloch et al., 1999; Sarwar et al., 2012; WMO, 2014; Kolesar et al., 2018). It is useful to define $Cl_y$ as total gas-phase inorganic chlorine, excluding particle phase $Cl^-$ . Most of this $Cl_y$ is present as HCl, which is removed rapidly by deposition but also serves as a source of chlorine radicals. Rapid cycling takes place between the chlorine radicals and other chlorine gases, eventually returning HCl. Thus it is useful to define reactive chlorine $Cl^*$ as the ensemble of $Cl_y$ gases other than HCl.

Cycling of chlorine affects tropospheric chemistry in a number of ways (Finlayson-Pitts, 2003; Saiz-Lopez and von Glasow, 2012). Acid displacement of $Cl^-$ by nitric acid ($HNO_3$) is a source of $NO_3^-$ aerosol (Massucci et al., 1999). Cl atoms provide a sink for methane, other volatile organic compounds (VOCs) (Atkinson, 1997), dimethyl sulfide (DMS) (Hoffmann et al., 2016; Chen et al., 2017), and mercury (Horowitz et al., 2017). Cycling between Cl radicals and their reservoirs drives catalytic ozone loss, and converts nitrogen oxide radicals ($NO_x \equiv NO + NO_2$) to $HNO_3$, decreasing both ozone and OH. On the other hand, aqueous-phase reaction of $Cl^-$ with $N_2O_5$ in polluted environments produces $ClNO_2$ radicals that photolyze in the daytime to return Cl atoms and $NO_2$, stimulating ozone production (Behnke et al., 1997; Osthoff et al., 2008). Chlorine also interacts with other halogens (bromine, iodine), initiating further radical chemistry that affects ozone, OH, and mercury.

A number of global modeling studies have investigated tropospheric halogen chemistry but most have focused on bromine and iodine, which are more active than chlorine because of the lower chemical stability of HBr and HI (Parrella et al., 2012; Sherwen et al., 2016a). Interest in global modeling of tropospheric chlorine has focused principally on quantifying the Cl atom concentration as a sink for methane (Keene et al., 1990; Singh et al., 1996). Previous global 3-D models found mean tropospheric Cl atom concentrations of the order of $10^3$ $cm^{-3}$, with values up to $10^4$ $cm^{-3}$ in the marine boundary layer (MBL), and contributing 2-3% of atmospheric methane oxidation (Long et al., 2014; Hossaini et al., 2016; Sherwen et al., 2016b; Schmidt et al., 2016). A regional modeling study by Sarwar et

al. (2014) included ClNO$_2$ chemistry in a standard ozone mechanism and found increases in surface ozone mixing ratios over the US of up to 7 ppb (ppb $\equiv$ nmol mol$^{-1}$).

Here we present a more comprehensive analysis of global tropospheric chlorine chemistry and its implications, building on previous model development of oxidant-aerosol-halogen chemistry in GEOS-Chem. A first capability for modeling tropospheric bromine in GEOS-Chem was introduced by Parrella et al. (2012). Eastham et al. (2014) extended it to describe stratospheric halogen chemistry including chlorine and bromine cycles. Schmidt et al. (2016) updated the tropospheric bromine simulation to include a broader suite of heterogeneous processes, and extended the Eastham et al. (2014) stratospheric chlorine scheme to the troposphere. Sherwen et al. (2016a;b) added iodine chemistry and made further updates to achieve a consistent representation of tropospheric chlorine-bromine-iodine chemistry in GEOS-Chem. Chen et al. (2017) added the aqueous-phase oxidation of SO$_2$ by HOBr and found a large effect on the MBL bromine budget. Our work advances the treatment of tropospheric chlorine in GEOS-Chem to include in particular a consistent treatment of SSA chloride and chlorine gases, SSA acid displacement thermodynamics, improved representation of heterogeneous chemistry, and better accounting of chlorine sources. We evaluate the model with a range of global observations for chlorine and related species. From there we quantify the global tropospheric chlorine budgets, describe the principal chemical pathways, and explore the impacts on tropospheric chemistry.

## 2 Model description

### 2.1 GEOS-Chem model with Cl+Br+I halogen chemistry

We build our new tropospheric chlorine simulation capability onto the standard version 11-02d of GEOS-Chem (http://www.geos-chem.org). The standard version includes a detailed tropospheric oxidant-aerosol-halogen mechanism as described by Sherwen et al. (2016b) and Chen et al. (2017). It includes 12 gas phase Cl$_y$ species: Cl, Cl$_2$, Cl$_2$O$_2$, ClNO$_2$, ClNO$_3$, ClO, ClOO, OClO, BrCl, ICl, HOCl, and HCl. It allows for heterogeneous chemistry initiated by SSA Cl$^-$ but does not actually track the SSA Cl$^-$ concentration and its exchange with gas-phase Cl$_y$. Here we add two new transported reactive species to GEOS-Chem to describe Cl$^-$ aerosol, one for the fine mode (< 1µm diameter) and one for the coarse mode (> 1µm diameter). The standard GEOS-Chem wet deposition schemes for water-soluble aerosols (Liu et al., 2001) and gases (Amos et al., 2012) are applied to Cl$^-$ aerosol and Cl$_y$ gases respectively, the latter with Henry's law constants from Sander (2015). Dry deposition of Cl$^-$ aerosol follows that of SSA (Jaeglé et al., 2011), and dry deposition of Cl$_y$ gases follows the resistance-in-series scheme of Wesely (1989) as implemented in GEOS-Chem by Wang et al. (1998). We also add to the model two SSA alkalinity tracers in the fine and coarse modes, and retain the inert SSA tracer to derive local concentrations of non-volatile SSA cations (Section 2.3). SSA debromination by oxidation of Br$^-$ as described by Schmidt et al. (2016) was included only as an option in standard version 11-02d of GEOS-Chem because of concern over excessive MBL BrO (Sherwen et al., 2016b). However, Zhu et al. (2018) show that it allows in fact a successful simulation of MBL BrO when one accounts for new losses in the standard model from aqueous-phase oxidation of SO$_2$ by HOBr (Chen et al., 2017) and oxidation of marine acetaldehyde by Br atoms (Millet et al., 2010). We include SSA debromination in this work.

We present a 1-year global simulation for 2016 driven by GEOS-FP (forward processing) assimilated meteorological
fields from the NASA Global Modeling and Assimilation office (GMAO) with native horizontal resolution of 0.25°
× 0.3125° and 72 vertical levels from the surface to the mesosphere. Our simulation is conducted at 4° × 5° horizontal
resolution and meteorological fields are conservatively regridded for that purpose. Stratospheric chemistry is
represented using 3-D monthly mean production rates and loss rate constants from a fully coupled stratosphere-
troposphere GEOS-Chem simulation (Murray et al., 2012; Eastham et al., 2014).
**2.2 Sources of chlorine**
Table 1 lists the global sources and sinks of tropospheric gas-phase inorganic chlorine ($Cl_y$) and reactive chlorine (Cl*)
in our model. The main source is mobilization of $Cl^-$ from SSA. SSA emission is computed locally in GEOS-Chem
separately for fine and coarse as the integrals of the size-dependent source function over two size bins, fine (0.2-1 μm
diameter) and coarse (1-8 μm diameter). The source function depends on wind speed and sea surface temperature
(Jaeglé et al., 2011). We obtain a global SSA source of 3230 Tg $a^{-1}$ for 2016, corresponding to 1780 Tg $a^{-1}$ $Cl^-$
(assuming fresh SSA to be 55.05% $Cl^-$ by mass (Lewis and Schwartz, 2013), of which 16% is in the fine mode and
84% is in the coarse mode. Only 64 Tg $Cl^-$ $a^{-1}$ (3.6%) is mobilized to $Cl_y$ by acid displacement and other heterogeneous
reactions, while the rest is deposited. 42% of the mobilization is from fine SSA and 58% is from coarse SSA. Details
of this mobilization are in Sections 2.3 and 2.4. 80% of the mobilization is by acid displacement to HCl, which is in
turn efficiently deposited. Only 19% of HCl is further mobilized to Cl* by reaction with OH to drive chlorine radical
chemistry. Direct generation of Cl* from SSA through heterogeneous chemistry provides a Cl* source of comparable
magnitude to HCl + OH, with dominant contributions from HOBr + $Cl^-$ and $N_2O_5$ + $Cl^-$ (the latter in polluted high-
$NO_x$ environments).
Cl* can also be produced in the model by atmospheric degradation of the organochlorine gases $CH_3Cl$, $CH_2Cl_2$, $CHCl_3$,
and $CH_2ICl$. These gases are mainly of biogenic marine origin, with the exception of $CH_2Cl_2$ which has a large
industrial solvent source (Simmonds et al., 2006). Mean tropospheric lifetimes are 520 days for $CH_3Cl$, 280 days for
$CH_2Cl_2$, 260 days for $CHCl_3$, and 0.4 days for $CH_2ICl$. Emissions of $CH_3Cl$, $CH_2Cl_2$, and $CHCl_3$ are implicitly treated
in the model by specifying monthly mean surface air boundary conditions in 5 latitude bands (60-90°N, 30-60°N, 0-
30°N, 0-30°S, and 30-90°S) from AGAGE observations (Prinn et al., 2018). Emission of $CH_2ICl$ is from Ordóñez et
al. (2012), as described by Sherwen et al. (2016a). Tropospheric oxidation of hydrochlorofluorocarbons (HCFCs) is
neglected as a source of Cl* because it is small compared to the other organochlorines.  The stratospheric source of
$Cl_y$ from chlorofluorocarbons (CFCs), HCFCs, and $CCl_4$ is included in the model on the basis of the Eastham et al.
(2014) GEOS-Chem stratospheric simulation as described in Section 2.1. Tropospheric organochlorines give a global
Cl* source of 3. 3 Tg Cl $a^{-1}$ in Table 1, smaller than that from heterogeneous SSA $Cl^-$ reactions (11.9 Tg Cl $a^{-1}$) or
oxidation of HCl by OH (9.7 Tg Cl $a^{-1}$). The stratosphere is a minor global source of tropospheric Cl* (0.06 Tg Cl $a^{-1}$
$^{1}$) although it could be important in the upper troposphere (Schmidt et al., 2016).
We also include primary HCl emissions from open fires. We apply the emission factors of (HCl + Cl⁻) from Lobert et
al. (1999) for different vegetation types to the GFED4 (Global Fire Emissions Database) biomass burned inventory
(van der Werf et al., 2010; Giglio et al., 2013), resulting in a global source of 7.6 Tg Cl a⁻¹ emitted as HCl.
Anthropogenic sources of HCl include coal combustion, waste incineration, and industrial activities. The only global
emission inventory is that of McCulloch et al. (1999), which gives a total of 6.7 Tg Cl a⁻¹. As shown in Section 4.2,
we find that this greatly overestimates atmospheric observations of HCl over the US. National inventories of HCl from
coal combustion available for China (236 Gg Cl a⁻¹ in 2012; (Liu et al., 2018)) and the US (69 Gg Cl a⁻¹ in 2014; (US
EPA, 2018)) are respectively six and seven times lower than McCulloch et al. (1999) for those countries. We choose
therefore not to include anthropogenic HCl emissions in our standard simulation, as they are small in any case from a
global budget perspective. We show in Section 4.2 that we can account for HCl observations in continental air largely
on the basis of the SSA Cl⁻ source. We also do not consider Cl* generation from snow/ice surfaces which could be
important in the Arctic spring MBL (Liao et al., 2014) but is highly uncertain and would only affect a small
atmospheric domain.
We do not include the anthropogenic Cl⁻ source from fugitive dust, although it might be important in contributing to
chloride levels in continental surface air (Sarwar et al., 2012). The global source of anthropogenic fugitive dust is
estimated to be less than 13 Tg a⁻¹ (Philip et al., 2017), of which 0.3% by mass is estimated to be chloride (Reff et al.,
2009). This corresponds to a chloride source of less than 0.39 Tg Cl a⁻¹, negligible on a global scale.
**2.3 HCl/Cl⁻ acid displacement thermodynamics**
SSA Cl⁻ can be displaced to HCl by strong acids (H₂SO₄, HNO₃) once the SSA is sufficiently aged that its initial
supply of alkalinity ($\equiv HCO_3^- + 2\times CO_3^{2-}$) has been exhausted. The acid displacement is described by
$Cl^- + HNO_3 \rightleftarrows HCl + NO_3^-$                (R1)
$Cl^- + H_2SO_4 \rightleftarrows HCl + HSO_4^-$           (R2)
with equilibrium constants from Fountoukis and Nenes, (2007). (R1) must be treated as an equilibrium because HNO₃
and HCl have comparable effective Henry's law constants. H₂SO₄ has a much lower vapor pressure so that (R2) fully
displaces HCl. Additional displacement of HCl by HSO₄⁻ does not take place because HSO₄⁻ is a much weaker acid
than HCl (Jacob et al., 1985).
Alkalinity initially prevents any acid displacement in freshly emitted SSA. Alkalinity is emitted as 0.07 mole
equivalents per kg of dry SSA (Gurciullo et al., 1999), and is transported in the model as two separate tracers for fine
and coarse SSA. It is consumed over time by uptake of acids (SO₂, H₂SO₄, HNO₃, and HCl) as described by Alexander
et al. (2005), and once fully consumed it is set to zero (titration). The SSA is then diagnosed as acidified, enabling
acid displacement by (R1)-(R2). In our simulation, alkalinity is titrated everywhere shortly after emission except in
some areas of the Southern Ocean, which is consistent with the model results of Alexander et al. (2005) and Kasibhatla
et al. (2018).
Observations in the MBL indicate that fine SSA is usually internally mixed with sulfate-nitrate-ammonium (SNA)
aerosols while coarse SSA is externally mixed (Fridlind and Jacobson, 2000; Dasgupta et al., 2007). Acid displacement
for the acidified fine SSA is thus computed by adding $HCl/Cl^-$ to the SNA thermodynamics. The local thermodynamic
gas-aerosol equilibrium for the resulting $H_2SO_4$-$HCl$-$HNO_3$-$NH_3$-NVC system is calculated with ISORROPIA II
(Fountoukis and Nenes, 2007). The calculation is done assuming an aqueous aerosol even if relative humidity is below
the deliquescence point (metastable state). NVC (non-volatile cations) describes the sum of cations emitted as SSA
and is treated in ISORROPIA II using $Na^+$ as proxy. Here NVC is emitted as 16.4 moles equivalent per kg of dry SSA
to balance the emission of SSA anions including $Cl^-$ , alkalinity, and sea-salt sulfate. The NVC concentration is
determined locally from the mass concentration of the inert SSA tracer.
Acid displacement for acidified coarse SSA is assumed to be driven by uptake of strong acids from the gas phase,
mainly $HNO_3$ (Kasibhatla et al., 2018). The ISORROPIA II calculation is conducted with 2 gas species ($HNO_3$ and
$HCl$) and 4 aerosol species (NVC, $Cl^-$, $SO_4^{2-}$, and $NO_3^-$). Here the sulfate includes only the emitted sea-salt component
and that produced by heterogeneous $SO_2$ oxidation in coarse SSA (Alexander et al., 2005). In the case of coarse
aerosols, there may be significant mass transfer limitation to reaching gas-aerosol thermodynamic partitioning (Meng
and Seinfeld, 1996). To account for this limitation, the concentrations are adjusted after the ISORROPIA II calculation
following the dynamic method of Pilinis et al. (2000). This 2-step thermodynamics approach has been used in previous
studies (Koo et al., 2003; Kelly et al., 2010).
**2.4 Heterogeneous chemistry of $Cl^-$**
Table 2 lists the heterogeneous reactions of $Cl^-$ other than acid displacement. The loss rate of a gas species X due to
reaction with $Cl^-$ is calculated following Jacob (2000):
$$\frac{dn_X}{dt} = -\left(\frac{r}{D_g} + \frac{4}{c\gamma([Cl^-])}\right)^{-1} An_X \qquad (1)$$
Here $n_X$ is the number density of species X (molecules of X per unit volume of air), $A$ is the aerosol or cloud surface
area concentration per unit volume of air, $r$ is the effective particle radius, $D_g$ is the gas-phase molecular diffusion
coefficient of X, $c$ is the average gas-phase thermal velocity of X, and $\gamma$ is the reactive uptake coefficient which is a
function of the aqueous-phase molar $Cl^-$ concentration $[Cl^-]$ (moles of $Cl^-$ per liter of water). Values of $\gamma$ in Table 2
are mostly from recommendations by the International Union of Pure and Applied Chemistry (IUPAC) (Ammann et
al., 2013).
The heterogeneous reactions take place in both clear-air aerosol and clouds. The GEOS-FP input meteorological data
include cloud fraction and liquid/ice water content for every grid cell. Concentrations per $cm^3$ of air of aerosol-phase
species (including fine and coarse $Cl^-$ and $Br^-$) within a grid cell are partitioned between clear air and cloud as
determined by the cloud fraction.  Clear-air aqueous-phase concentrations for use in calculating heterogeneous
reaction rates are derived from the RH-dependent liquid water contents of fine and coarse SSA using aerosol
hygroscopic growth factors from the Global Aerosol Database (GADS, (Koepke, 1997)) with update by Lewis and
Schwartz (2006). In-cloud aqueous-phase concentrations are derived using liquid and ice water content from the
GEOS-FP meteorological data.  Values of $r$ in equation (1) are specified as RH-dependent effective radii for the
different clear-air aerosol components (Martin et al., 2003), and are set to 10 µm for cloud droplets and 75 µm for ice
particles. These effective radii are also used to infer the area concentrations $A$ on the basis of the mass concentrations.
Heterogeneous chemistry in ice clouds is restricted to the unfrozen layer coating the ice crystal, which is assumed to
be 1% of the ice crystal radius (Schmidt et al., 2016).
The reactions of HOBr, HOCl, and $ClNO_2$ with $Cl^-$ in Table 2 are pH-dependent and require acidic conditions (Fickert
et al., 1999; Abbatt et al., 2012). They are considered only when SSA alkalinity has been titrated, and $ClNO_2$ + $Cl^-$
further requires pH <2. The pH of chloride-containing fine aerosol after alkalinity has been titrated is calculated by
ISORROPIA II. Liquid cloud water pH is calculated in GEOS-Chem following Alexander et al. (2012), with update
to include $Cl^-$ and NVC. Coarse-mode SSA and ice cloud pH are assumed to be 5 and 4.5 respectively (Schmidt et al.,

12   2016).

**3 Global budget and distribution of tropospheric chlorine**
Figure 1 describes the global budget and cycling of tropospheric inorganic chlorine in GEOS-Chem. The dominant
source of $Cl_y$ is acid displacement from SSA.  The global rate of $Cl_y$ generation from acid displacement is 52 Tg Cl a$^-$
$^1$, close to the observationally based estimate of 50 Tg Cl a$^{-1}$ by Graedel and Keene (1995), and lower than the model
estimate of 90 Tg Cl a$^{-1}$ from Hossaini et al. (2016), who treated displacement of $Cl^-$ by $HNO_3$ as an irreversible rather
than thermodynamic equilibrium process. HCl is the largest reservoir of $Cl_y$ in the troposphere, with a global mean
tropospheric mixing ratio of 60 ppt (ppt $\equiv$ pmol/mol).
Acid displacement generates $Cl_y$ as HCl, which is mostly removed by deposition. Broader effects of chlorine on
tropospheric chemistry take place through the cycling of radicals originating from production of reactive chlorine Cl*
$\equiv Cl_y$ – HCl. HCl contributes 9.7 Tg Cl a$^{-1}$ to Cl* through the reaction between HCl and OH. Beside this source, $Cl^-$
provides a Cl* source of 12 Tg Cl a$^{-1}$ through heterogeneous reactions with principal contributions from HOBr + $Cl^-$
(8.6 Tg Cl a$^{-1}$) and $N_2O_5$ + $Cl^-$ (1.8 Tg Cl a$^{-1}$). This heterogeneous source of 12 Tg Cl a$^{-1}$ is much higher than previous
estimates of 5.6 Tg Cl a$^{-1}$ (Hossaini et al., 2016) and 6.1 Tg Cl a$^{-1}$ (Schmidt et al., 2016). Schmidt et al. (2016) only
considered the HOBr + $Cl^-$ reaction.  Production of the chlorine radicals Cl and ClO is contributed by the HCl + OH
reaction (45%) and by photolysis of BrCl (40%), $ClNO_2$ (8%), $Cl_2$ (4%), and ICl (2%). Loss of Cl* is mainly through
the reaction of Cl with methane (46%) and other organic compounds ($CH_3OH$ 15%, $CH_3OOH$ 11%, $C_2H_6$ 8%, higher
alkanes 8%, and $CH_2O$ 7%).
Conversion of Cl to ClO* drives some cycling of chlorine radicals, but the associated chain length versus Cl* loss is
short ($4.1 \times 10^4 / 2.5 \times 10^4 = 1.6$). Conversion of Cl to ClO is mainly by reaction with ozone (98%), while conversion of
ClO back to Cl is mostly by reaction with NO (72%), driving a null cycle as $NO_2$ photolyzes to regenerate NO and
ozone.

Figure 2 shows the annual mean global distributions of HCl mixing ratios and Cl atom concentrations. The mixing ratio of HCl decreases from the surface to the middle troposphere, reflecting the SSA source, and then increases again in the upper troposphere where it is supplied by transport from the stratosphere and has a long lifetime due to lack of scavenging. Remarkably, Cl atom number densities show little decrease with altitude, contrary to the common assumption that tropospheric Cl atoms should be mainly confined to the MBL where the SSA source resides (Singh et al., 1996). We find that the effect of the SSA source is offset by the slower sink of Cl* at higher altitudes due to the strong temperature dependence of the reactions between Cl atom and organic compounds. Transport of HCl and Cl* from the stratosphere also contribute to the source of Cl atoms in the upper troposphere.

HCl mixing ratios in marine surface air are usually highest along polluted coastlines where the large sources of $HNO_3$ and $H_2SO_4$ from anthropogenic $NO_x$ and $SO_2$ emissions drive acid displacement from SSA. By contrast, HCl mixing ratios over the Southern Ocean are low because of the low supply of acid gases. The distribution of Cl atoms in surface air reflects its sources from both HCl + OH and the heterogeneous production of Cl*. The highest concentrations are in northern Europe due to production of $ClNO_2$ from the $N_2O_5$ + Cl$^-$ reaction (R3). Cl atom concentrations in marine air are shifted poleward relative to HCl because of increasing bromine radical concentrations (Parrella et al., 2012), driving BrCl formation by the HOBr + Cl$^-$ reaction (R5).

Figure 3 shows the global mean vertical distributions of reactive chlorine species (Cl*) in continental and marine air. Mean boundary layer mixing ratios are higher over land than over the ocean because of the $ClNO_2$ source from $N_2O_5$ +Cl$^-$ in high-$NO_x$ polluted air (Thornton et al., 2010). $ClNO_2$ mixing ratios are much higher than in the Sherwen et al. (2016b) model which restricted its production to SSA, reflecting the importance of HCl dissolved in SNA aerosol which allows further transport inland. High mixing ratios of $ClNO_3$ in the upper troposphere are due to transport from the stratosphere and inefficacy of the sinks from hydrolysis and heterogeneous chemistry. In the marine MBL we find comparable contributions from HOCl (mainly in daytime) and $Cl_2$ and $ClNO_2$ (mainly at night). The BrCl mixing ratio is much lower than in the previous model studies of Long et al. (2014) and Sherwen et al. (2016b) which had very large sources from the HOBr + Cl$^-$ reaction (R5). Our lower BrCl mixing ratio is due to competition from the HOBr + S(IV) reaction (Chen et al., 2017) and to oceanic VOC emissions (Millet et al., 2010), both of which act to depress bromine radical concentrations in the MBL (Zhu et al., 2018). Further discussion of the BrCl source is presented in Section 5.2.

**4 Comparison to observations**

Here we compare the model simulation for 2016 to observations for gas-phase chlorine and related species collected in different years, assuming interannual variability to be a minor factor in model error. Previous evaluation of the GEOS-Chem sea salt source by Jaeglé et al. (2011) showed general skill in simulating SSA observations and we do not repeat this evaluation here. We also do not consider data affected by local anthropogenic sources because they would not be properly resolved at the 4°×5° grid resolution of our model.

**4.1 Surface air observations**

Table 3 compares our simulated $Cl^-$ SSA deficits to an ensemble of marine air observations compiled by Graedel and Keene (1995). The $Cl^-$ deficit is relative to seawater composition and provides an indicator of the mobilization of $Cl^-$ through acid displacement and heterogeneous chemistry. The observations show a wide range from -50% to +90%, and Graedel and Keene (1995) emphasize that uncertainties are large. Slight negative deficits in the observations could be caused by titration of alkalinity by HCl but large negative deficits are likely due to error. Mean model deficits sampled for the regions and months of the observations range from +4% to +40%, not inconsistent with the observations. The largest model deficits are in polluted coastal regions because of acid displacement and this is also where the measured deficits are largest.

Figure 4 compares simulated HCl and $HNO_3$ mixing ratios to concurrent observations of both gases at coastal sites and over oceans. The data are arranged from left to right by increasing latitude. Mean HCl mixing ratios average 323 ppt in the model and 347 ppt in the observations for the ensemble of regions. The HCl source in the model is mainly acid displacement from SSA. A sensitivity simulation without acid displacement from SSA has less than 7 ppt HCl in all regions. The model captures the spatial variability of the mean HCl mixing ratios across locations ($r = 0.88$), which largely reflects the HCl enhancement at polluted coastal sites and northern mid-latitudes (Figure 2). Simulated $HNO_3$ mixing ratios average 190 ppt across locations as compared to 137 ppt in the observations, again with good simulation of spatial variability ($r = 0.96$) driven by $NO_x$ emissions. $HNO_3$ mixing ratios are sensitive to acid displacement from SSA, as the sensitivity simulation without acid displacement shows mean values of 441 ppt that are much higher than observed. This could partly explain the general model problem of overestimating $HNO_3$ in remote air (Bey et al., 2001).

Figure 5 shows 2016 annual mean observations of $PM_{2.5}$ $Cl^-$ (mass concentration in particles less than 2.5 μm diameter) from the US Interagency Monitoring of Protected Visual Environments (IMPROVE) network (Malm et al., 1994). Corresponding model values are shown as background contours for fine $Cl^-$ (< 1μm diameter and internally mixed with SNA aerosol) are for total $Cl^-$. One would expect the IMPROVE concentrations to be higher than the model fine $Cl^-$ (because of the larger size cut) and lower than total $Cl^-$, and this is generally the case. . The model is consistent with observations in the continental interior, which we attribute to inland transport of marine HCl incorporated into SNA aerosol. Fine $Cl^-$ concentrations can actually be higher over the continent than over the ocean because of HCl displacement from the coarse SSA followed by re-condensation on anthropogenic SNA aerosol. The model underestimates the observations over the Southwest US and this may be due to a missing dust source. We find IMPROVE $Cl^-$ and dust concentrations are moderately to highly correlated (R = 0.3-0.9) at the sites in this region.

A number of surface air measurements have been made of $Cl^*$ as the water-soluble component of $Cl_y$ after removal of HCl (Keene et al., 1990), although most of these measurements are below the detection limit (Table 4, mostly from Keene et al. (2009)). This $Cl^*$ has been commonly assumed to represent the sum of $Cl_2$ and HOCl (Pszenny et al., 1993) but it would also include $ClNO_2$, $ClNO_3$, and minor components of $Cl^*$. Table 4 shows that simulated $Cl^*$ mixing

ratios are consistent with the measurements to the extent that comparison is possible. Simulated Cl* over remote
oceans is dominated by HOCl, but $ClNO_2$ is responsible for the high values over the Atlantic Ocean near Europe.
Lawler et al. (2009) measured $Cl_2$ and BrCl mixing ratios at Cape Verde in the tropical Atlantic  for 5 days in May-
June 2007, and Lawler et al. (2011) measured $Cl_2$ and HOCl mixing ratios at the same site for 7 days in May-June
2009.  The observations show a diurnal cycle with mixing ratios of $Cl_2$ highest at night, and HOCl highest in the day,
consistent with the model. Observed mixing ratios in background marine air were in the range 0-30 ppt for $Cl_2$ and 0-
2 ppt for BrCl at night, and 0-5 ppt for HOCl in the daytime. Corresponding mean model values are 0.3 ppt for $Cl_2$,
1.8 ppt for BrCl, and 5 ppt for HOCl, with little day-to-day variability. Lawler et al. (2011) also sampled long-range
outflow from Europe for 3 days in 2009 with daytime HOCl and nighttime $Cl_2$ mixing ratio ranges of 40-200 ppt and
5-40 ppt respectively but the model does not capture these enhancements. Sommariva and von Glasow (2012)
suggested that a lower aerosol pH and/or slower rate for $HOCl + Cl^-$ could explain the high HOCl in European outflow
but this would also cause $Cl_2$ to be lower. We have no explanation for the high $Cl_2$ values observed by Lawler et al.
(2009; 2011) in marine air or for the joint observed enhancements of HOCl and $Cl_2$ in European outflow.
Many surface observations of $ClNO_2$ have been made in nighttime urban environments. These are difficult to compare
to the model because of the $4^o \times 5^o$ grid resolution and because of nighttime stratification of the surface layer (the
lowest model grid level extends up to 130 m above the surface). In addition, the publications usually report maxima
instead of means. Table 5 shows a comparison for representative sites, indicating that the model offers a credible
simulation within the above caveats. The previous GEOS-Chem simulation of Sherwen et al. (2016b) only considered
$ClNO_2$ production in SSA and as a result their $ClNO_2$ mixing ratios were consistently below a few ppt at continental
sites. Our simulation can reproduce the observed >100 ppt concentrations at these sites because it accounts for HCl
dissolved in SNA aerosol, allowing marine influence to extend further inland as also shown in the comparison to the
IMPROVE $Cl^-$ data (Figure 5).
**4.2 Comparison to aircraft measurements**
The WINTER aircraft campaign over the eastern US and offshore in February-March 2015 provides a unique data set
for evaluating our model. Measurements included HCl, $ClNO_2$, HOCl, $Cl_2$, and $ClNO_3$ by Time of Flight Chemical
Ionization Mass Spectrometry (TOF-CIMS) (Lee et al., 2018). We focus on the first four measurements because
calibration for $ClNO_3$ needs further examination. The mean 1s detection limits for HCl, $ClNO_2$, HOCl, and $Cl_2$ were
100, 2, 2, and 1 ppt respectively (Lee et al., 2018). The estimated calibration uncertainty is ±30% for all chlorine
species. As discussed in Lee et al. (2018), labeled 15-$N_2O_5$ was added to the inlet tip during WINTER flights to
quantify inlet production of $ClNO_2$, which was found to be negligible (<<10% of measured $ClNO_2$), but inlet
production of $Cl_2$, for example, from surface reactions of HOCl with adsorbed HCl, was not evaluated.
Figure 6 compares the observed median vertical profiles of HCl, $ClNO_2$, HOCl, and $Cl_2$ in WINTER to the model
sampled along the flight tracks for the corresponding period. Figure 7 compares the median diurnal variations below
1 km altitude, separately over ocean and land. We exclude daytime (10:00-16:00 local) data for $ClNO_2$ in Figure 6
because its mixing ratios are near-zero (Figure 7).
The WINTER observations of HCl show median values of 380 ppt near the surface, dropping to a background of 100-
200 ppt in the free troposphere (Figure 6). The model is lower than the observations in the lowest 2 km but within the
calibration uncertainty. The free tropospheric background in the model is much lower than observed but the
observations are near the 100 ppt detection limit.  HCl mixing ratios in the lowest km average 60% higher over ocean
than over land in both the observations and the model, reflecting the marine source.
Also shown in Figure 6 is a sensitivity simulation including anthropogenic HCl emissions from McCulloch et al. (1999)
as described in Section 2.2. The resulting model mixing ratios are too high though still within the calibration
uncertainty. Based on sampling of power plant plumes during WINTER campaign, Lee et al. (2018) inferred a
HCl:$SO_2$ emission mass ratio of 0.033 from power plants. Adding this emission to the standard simulation, scaled to
the $SO_2$ emissions in GEOS-Chem (from the EPA National Emission Inventory over the US) increases modeled
mixing ratios of HCl in the continental boundary layer by 18% along the WINTER flight tracks, improving agreement
with observations relative to the standard simulation but still representing a relatively minor source. $ClNO_2$, HOCl,
and $Cl_2$ mixing ratios increase by12%, 8%, and 4% respectively.
Figure 8 compares the model HCl vertical profiles to measurements by the Georgia Tech CIMS instrument during the
SEAC[4]RS campaign over the Southeast US in August-September 2013 (Toon et al., 2016) and the KORUS-AQ
campaign over and around the Korea peninsula in May-June 2015. The standard model simulation without
anthropogenic chlorine successfully simulates the boundary layer HCl observations, but adding the McCulloch et al.
(1999) anthropogenic inventory results in large overestimates. Boundary layer HCl mixing ratios over land are much
lower in SEAC[4]RS than in WINTER and this is well reproduced by the model, where the difference is due to seasonal
contrast in the SSA source and in the inflow of marine air.  The free tropospheric background observed in SEAC[4]RS
and KORUS-AQ data is only ~25 ppt, much lower than in WINTER (100-200 ppt), whereas the model free
tropospheric background is consistently 20-50 ppt in all three campaigns. The WINTER observations are near their
100 ppt detection limit as pointed out above.
Mixing ratios of $ClNO_2$ observed in WINTER are above the detection limit only in the lowest km of atmosphere at
night, and are much higher over the ocean than over land.  This is well simulated by the model (Figures 6 and 7), and
reflects the nighttime source from the $N_2O_5$ + $Cl^-$ heterogeneous reaction from the combined with the fast loss by
photolysis in daytime. Our results contrast with previous studies suggesting that the Bertram and Thornton (2009)
representation of $ClNO_2$ production from the $N_2O_5$ + $Cl^-$ heterogeneous reaction (Table 2) results in an overestimate
of $ClNO_2$ observations (Riedel et al., 2013; Wagner et al., 2013; McDuffie et al., 2018a;b). By using a box model
applied to the WINTER observations, McDuffie et al. (2018a;b) found that both $N_2O_5$ uptake rate and $ClNO_2$
production yield were overestimated by the Bertram and Thornton (2009) parameterization. One important difference
with our simulation is the assumption of aerosol mixing state. When computing $N_2O_5$ reactive uptake with the
parameterization of Table 2, McDuffie et al. (2018a;b) assumed $Cl^-$ to be internally mixed across all aerosol types
(including in particular organic aerosol). In contrast, GEOS-Chem assumes that Cl⁻ is present only in SNA and SSA
when doing the calculation of $N_2O_5$ reactive uptake rates, assuming an external mixture of aerosol types (Martin et al.,
2003; Evans and Jacob, 2005). This decreases both the $N_2O_5$ uptake rate and the $ClNO_2$ yield as compared to the
internal mixing assumption of McDuffie et al. (2018a;b), although it is not clear which assumption is best.
Nighttime $Cl_2$ mixing ratios in WINTER are greatly overestimated by the model. Under polluted wintertime conditions
such as in WINTER the $ClNO_2$ + Cl⁻ reaction greatly enhances $Cl_2$ production in the model:
$$ClNO_2 + Cl^- \rightarrow NO_2^- + Cl_2 \qquad\qquad (R7)$$
The reactive uptake coefficient for (R7) in Table 2 is based on a single laboratory study (Roberts et al., 2008). It
requires an aerosol pH < 2 and this condition is generally met for our model simulation of the WINTER environment,
consistent with the observation-based analysis of aerosol pH by Guo et al. (2016) for the eastern US in winter. A
sensitivity simulation without (R7) is shown as dashed red lines in Figure 7 and can reproduce the low $Cl_2$ mixing
ratios observed over the ocean at night. The analysis of WINTER data by McDuffie et al. (2018b) finds that the
correlation between particle acidity and $Cl_2$ observations is opposite of the trend expected from (R7). Further study of
that reaction is needed.
The model underestimates the WINTER observations of HOCl and $Cl_2$ in daytime, over the ocean as well as over land.
These species have short lifetimes against photolysis (less than a few minutes). Direct anthropogenic emission from
coal combustion has been proposed (Chang et al., 2002) but would only be observed in plumes and not over the oceans.
Matching the >1 ppt $Cl_2$ observed during daytime is particularly problematic since it would require a large
photochemical source absent from the model. Lawler et al. (2011) suggested a fast daytime HOCl source from a
hypothetical light-dependent Cl⁻ oxidation. The measurements of $Cl_2$ are also possibly subject to positive artifact from
rapid heterogeneous conversion of chlorine species on the surface of the TOF-CIMS inlet (Lee et al., 2018).
**5 Global implications of tropospheric chlorine chemistry**
**5.1 Cl atom and its impact on VOCs**
The global mean pressure-weighted tropospheric Cl atom concentration in our simulation is 620 cm⁻³, while the MBL
concentration averages 1200 cm⁻³ (Figure 2). Our global mean is lower than the previous global model studies of
Hossaini et al. (2016) (1300 cm⁻³) and Long et al. (2014) (3000 cm⁻³), which had excessive Cl* generation as discussed
above. It is consistent with the upper limit of 1000 cm⁻³ inferred by Singh et al. (1996) from global modeling of $C_2Cl_4$
observations ($C_2Cl_4$ is highly reactive with Cl atoms). Isotopic observations of methane have been used to infer a Cl
atom concentration in the MBL higher than 9000 cm⁻³ in the extra-tropical Southern Hemisphere (Platt et al., 2004;
Allan et al., 2007), much higher than our estimate of 800 cm⁻³ over this region. More recently, Gromov et al. (2018)
revisited these data together with added constraints from CO isotope measurements and concluded that extra-tropical
Southern Hemisphere concentrations of Cl atoms in the MBL should be lower than 900 $cm^{-3}$, consistent with our
estimate.
Tropospheric oxidation by Cl atoms drives a present-day methane loss rate of 5.3 Tg $a^{-1}$ in our model, contributing
only 1.0% of total methane chemical loss. It has more significant impact on the oxidation of some other VOCs,
contributing 20% of the global loss for ethane, 14% for propane, 10% for higher alkanes, and 4% for methanol.
**5.2 Impact on bromine and iodine chemistry**
Bromine radicals ($BrO_x \equiv Br + BrO$) and iodine radicals ($IO_x \equiv I + IO$) affect global tropospheric chemistry by
depleting ozone and OH (Parrella et al., 2012; Sherwen et al., 2016b). Br atoms are also thought to drive the oxidation
of elemental mercury (Holmes et al., 2006). Chlorine chemistry increases $IO_x$ mixing ratios by 16% due to the
reactions of HOI, $INO_2$, and $INO_3$ with $Cl^-$ (R11-R13), producing ICl which photolyzes rapidly to I atoms (Figure 1).
The effect on bromine is more complicated. Bromine radicals originate from photolysis and oxidation of
organobromines emitted by the ocean, as well as from SSA debromination (Yang et al., 2005). They are lost by
conversion to HBr which is efficiently deposited. Parrella et al. (2012) pointed out that heterogeneous chemistry of
HBr (dissolved as $Br^-$) is critical for recycling bromine radicals and explaining observed tropospheric BrO mixing
ratios in the background troposphere:
$HOBr(aq) + Br^- + H^+ \rightarrow Br_2 + H_2O$ (R14)
$Br_2 + hv \rightarrow 2Br$ (R15)
Chloride ions and dissolved $SO_2$ ($S(IV) \equiv HSO_3^- + SO_3^{2-}$) can however compete with $Br^-$ for the available HOBr (Chen
et al., 2017):
$HOBr(aq) + Cl^- + H^+ \rightarrow BrCl + H_2O$ (R5)
$HOBr(aq) + HSO_3^-/SO_3^{2-} \rightarrow HBr + HSO_4^-/SO_4^{2-}$ (R16)
Chen et al. (2017) pointed out that reaction (R16) effectively decreases BrO mixing ratios by producing HBr which is
rapidly deposited instead of contributing to $BrO_x$ cycling. They found in a GEOS-Chem simulation that global
tropospheric BrO mixing ratios decreased by a factor of 2 as a result. Reaction (R5) may however have a compensating
or opposite effect.  It propagates the cycling of $BrO_x$ if BrCl volatilizes:
$BrCl + hv \rightarrow Br + Cl$ (R17)
but it may also generate new $BrO_x$ if BrCl reacts with $Br^-$ in the aqueous phase to produce $Br_2$ (Wang et al., 1994):
$BrCl(aq) + Br^- \rightleftarrows Br_2Cl^-$ (R18)
$Br_2Cl^- \rightleftarrows Br_2(aq) + Cl^-$ (R19)
The sequence (R5) + (R18) + (R19) with $Cl^-$ as a catalyst has the same stoichiometry as (R14) and thus contributes to
HBr recycling in the same way. We find in the model that it is globally 30 times faster than (R14) and therefore much
more effective at regenerating bromine radicals. In GEOS-Chem, the rate of reaction (R5) computed from Table 2 is
applied to the following stoichiometry reflecting the ensemble of reactions (R5, R14, R18, and R19):
$HOBr(aq) + YBr^- + (1 - Y)Cl^- + H^+ \rightarrow YBr_2 + (1 - Y)BrCl + H_2O$          (R5+R14+R18+R19)
where $Y$ is the yield of $Br_2$ and 1-$Y$ is the yield of BrCl. $Y$ is calculated following the laboratory study of Fickert et al.

5  (1999):

$Y = 0.41\log_{10}([Br^-]/[Cl^-]) + 2.25$          for $[Br^-]/[Cl^-] < 5 \times 10^{-4}$          (2)
$Y = 0.90$          for $[Br^-]/[Cl^-] > 5 \times 10^{-4}$          (3)
This mechanism was first included in GEOS-Chem version 11-02d by Chen et al. (2017), who did not however have
an explicit SSA Cl$^-$ simulation (they instead assumed a fixed SSA [Cl$^-$] = 0.5 M, and considered only dissolved HCl
in cloud).
Chen et al. (2017) found in their GEOS-Chem simulation that the global tropospheric BrO burden was 8.7 Gg without
the HOBr + S(IV) reaction (R16), and dropped to 3.6 Gg when the reaction was included. Previous GEOS-Chem
model estimates of the global tropospheric BrO burden were 3.8 Gg (Parrella et al., 2012), 5.7 Gg (Schmidt et al.,
2016), and 6.4 Gg (Sherwen et al., 2016b). Our simulation features many updates relative to Chen et al. (2017)
including not only explicit SSA Cl$^-$ but also explicit calculation of aerosol pH with ISORROPIA II for the rates of
reactions in Table 2. By including explicit SSA Cl$^-$, the cloudwater [Cl$^-$] in our model is much higher than that in Chen
et al. (2017) and more comparable to measurements (~10$^{-4}$ M in typical cloud; (Straub et al., 2007)). We find in our
standard simulation a global tropospheric BrO burden of 4.2 Gg, 17% higher than Chen et al. (2017).
Figure 9 shows the change of surface BrO mixing ratios due specifically to tropospheric chlorine chemistry, as
obtained by difference with a sensitivity simulation including none of the Cl$_y$ chemistry shown in Figure 1. The
inclusion of chlorine chemistry increases the global tropospheric BrO burden by 85%. More than 80% of this change
is caused by the HOBr + Cl$^-$ reaction as discussed above. Other significant contributions include ClNO$_3$ + Br$^-$ and
ClNO$_2$ + Br$^-$. The largest BrO increases (1-2 ppt) are in surface air over the high northern latitudes oceans where SSA
emissions are high and acidic conditions promote HOBr + Cl$^-$ chemistry.
**5.3 Impact on tropospheric ozone and OH**
Figure 9 also shows the effects of chlorine chemistry on NO$_x$, OH, and ozone. The global tropospheric burdens
decrease by 5% for NO$_x$, 3% for OH, and 7% for ozone. The inter-hemispheric (N/S) ratio of tropospheric mean OH
decreases from 1.14 to 1.12. Models tend to overestimate global mean tropospheric OH and its inter-hemispheric ratio
relative to the constraint from methylchloroform which suggests a ratio of 0.85-0.98 (Naik et al., 2013;Voulgarakis et
al., 2013). The effect of chlorine chemistry on the N/S ratio is slight but in the right direction.

The chlorine-induced decreases in Figure 9 are mainly through bromine chemistry initiated by chlorine (Section 5.2), and have spatial distributions characteristic of bromine chemistry with maxima at high latitudes as discussed by Schmidt et al. (2016). There are specific chlorine mechanisms including catalytic ozone loss through HOCl formation and photolysis:

$$Cl + O_3 \rightarrow ClO + O_2 \qquad\qquad (R20)$$

$$ClO + HO_2 \rightarrow HOCl + O_2 \qquad\qquad (R21)$$

$$HOCl \overset{hv}{\rightarrow} Cl + OH \qquad\qquad (R22)$$

$$Net: \; O_3 + HO_2 \rightarrow OH + 2O_2$$

and also loss of $NO_x$:

$$ClO + NO_2 + M \; \rightarrow ClNO_3 + M \qquad\qquad (R23)$$

$$ClNO_3 + H_2O \rightarrow HOCl + HNO_3 \qquad\qquad (R24)$$

However, we find that the rates are very small compared to similar mechanisms involving bromine and iodine because the stability of HCl quenches Cl* radical cycling.

A particular situation arises over polluted continents due to $ClNO_2$ chemistry. Production of $ClNO_2$ at night from the $N_2O_5 + Cl^-$ heterogeneous reaction, followed by photolysis in the morning to release Cl and $NO_2$, provides a source of radicals and ozone. This explains the increases of OH over North America and Europe in Figure 9. The effect is most important at high northern latitudes in winter due to the longer night. To isolate the impact on ozone we conducted a sensitivity simulation with no $ClNO_2$ production, setting $\varphi = 0$ for reaction (R3) in Table 2. The surface air ozone enhancement due to $ClNO_2$ chemistry is found to be the largest (~8 ppb) in European winter, due to the large supply of $Cl^-$ from the North Atlantic combined with high $NO_x$ emissions. Other polluted continents see ozone increases of 1-5 ppb in winter. The effect in summer is less than 1 ppb. These results are similar to previous regional modeling studies by Sarwar et al. (2014) and Sherwen et al. (2017).

Figure S1 shows the differences of BrO, $NO_x$, OH, and ozone concentrations between our model and the standard GEOS-Chem model version 11-02d including SSA debromination. Our explicit treatment of chlorine chemistry and thermodynamic representation of aerosol pH increases the global tropospheric BrO burden by 40%. Most of this change is caused by faster $HOBr + Cl^-$ reaction at high latitudes, resulting from higher Cl- concentration in our model particularly in cloud. The decrease of BrO in the tropical MBL is caused by an increase in aerosol pH (pH was previously assumed to be 0 for computation of bromine chemistry), which slows down the acid-catalyzed recycling of bromine by reactions (R5) and (R14). Our computed global tropospheric burdens decrease by 4% for $NO_x$, 2% for OH, and 4% for ozone relative to version 11-02d, again due to the more active bromine chemistry. The increase of OH over continental regions is due to our accounting of HCl dissolved in SNA aerosol, allowing marine influence to extend further inland to drive $ClNO_2$ chemistry.

**6 Conclusions**

We have added to the GEOS-Chem model a comprehensive and consistent representation of tropospheric chlorine chemistry. This includes in particular explicit accounting of the mobilization of sea salt aerosol (SSA) chloride ($Cl^-$), by acid displacement of HCl as well as by other heterogeneous processes. Cycling of inorganic gas-phase chlorine species ($Cl_y$) generated from SSA and other sources is simulated and coupled to the model aerosol-oxidant-bromine-iodine chemistry. With our work, GEOS-Chem now has a complete simulation of halogen (Cl+Br+I) chemistry in both the troposphere and stratosphere.

Emission of chlorine in the model is mainly as sea-salt aerosol (1780 Tg Cl $a^{-1}$). Other sources (combustion, organochlorines, stratospheric input) are also included but are small in comparison. Most of the sea-salt aerosol chloride is removed by deposition, but 3.6% is mobilized to inorganic gas-phase chlorine ($Cl_y$) through acid displacement to HCl (52 Tg $a^{-1}$) and through other heterogeneous chemistry producing more reactive chlorine species (12 Tg $a^{-1}$). We define reactive chlorine (Cl*) as the ensemble of $Cl_y$ species excluding HCl and including Cl, ClO, $Cl_2$, BrCl, HOCl, $ClNO_2$, $ClNO_3$, plus other minor species. Oxidation of HCl by OH provides a Cl* source of 9.7 Tg $a^{-1}$, comparable to the heterogeneous source from HOBr + $Cl^-$ (8.6 Tg $a^{-1}$). $N_2O_5$ + $Cl^-$ (1.8 Tg $a^{-1}$) is also important in polluted environments. Cycling between Cl* species drives radical chlorine (Cl/ClO) chemistry but chain lengths are limited by fast conversion to HCl and subsequent deposition.

HCl mixing ratios in the model are highest over the oceans downwind of polluted continents due to effective acid displacement from sea-salt aerosol by $HNO_3$ and $H_2SO_4$. Mixing ratios are much lower over the Southern Ocean where the supply of acids is low. The dominant daytime Cl* species is generally HOCl while BrCl, $Cl_2$, and $ClNO_2$ dominate at night. $ClNO_3$ dominates in the upper troposphere due to stratospheric input. Chlorine atom concentrations are highest over Europe in winter due to $ClNO_2$ chemistry, and are otherwise high over the northern mid-latitudes oceans where the supply of acidity promotes Cl formation both through HCl and through acid-catalyzed heterogeneous processes.

Comparison of model results to observations in marine surface air show that the model is usually able to reproduce the range and distributions of observed sea-salt aerosol chloride deficits, HCl mixing ratios, and Cl* mixing ratios. In particular, concurrent observations of HCl and $HNO_3$ in coastal/marine air worldwide show high correlation with the model including high HCl mixing ratios at northern mid-latitudes combined with depressed $HNO_3$. Consideration of acid displacement greatly improves model agreement with $HNO_3$ observations in marine air. The model can also successfully simulate observations of high $ClNO_2$ at night including in continental air. The chlorine in that case originates from sea-salt aerosol transported far inland following uptake of volatilized HCl by sulfate-nitrate-ammonium (SNA) aerosol. The model cannot reproduce the very high HOCl and $Cl_2$ concentrations observed by Lawler et al. (2009;2011) at Cape Verde in the tropical Atlantic.

Comparisons of model results to aircraft campaign observations from WINTER (eastern US and offshore, Feb-Mar 2015), SEAC4RS (Southeast US, Aug-Sep 2013), and KORUS-AQ (Korean Peninsula, April-Jun 2016) show general

consistency for HCl vertical profiles. Continental boundary layer HCl mixing ratios in these campaigns can be mostly accounted for by the marine source transported inland, though power plants could make a minor contribution. WINTER observations also include $ClNO_2$ and $Cl_2$, and HOCl. The observed $ClNO_2$ is mainly confined to the nighttime marine boundary layer and is consistent with the model. Observed $Cl_2$ concentrations at night are much lower than the model, which has a large source under the WINTER conditions from the $ClNO_2 + Cl^-$ heterogeneous reaction. The rate coefficient for this reaction is from only one laboratory study.

The model simulates a global mean Cl atom concentration of 620 $cm^{-3}$ in the troposphere and 1200 $cm^{-3}$ in the marine boundary layer (MBL), lower than previous global model studies that had excessive generation of Cl* but consistent with independent proxy constraints. We find that oxidation by Cl atoms accounts for only 1.0% of the global loss of atmospheric methane but has larger effects on the global losses of ethane (20%), propane (14%), and methanol (4%). Chlorine chemistry increases global tropospheric BrO by 85%, and decreases ozone and OH by 7% and 3% respectively, relative to a sensitivity simulation with no chlorine chemistry. The large effect on BrO is due to production of bromine radicals by the $HOBr + Cl^-$ heterogeneous reaction, and the decreases of ozone and OH are mainly through the induced bromine chemistry. An exception is winter conditions over polluted regions, where $ClNO_2$ chemistry increases ozone mixing ratios by up to 8 ppb.

**Author contributions.** XW, DJJ, and HL designed the study. XW developed the chlorine model code, performed the simulations and analyses. SDE, MPS, LZ, QC, BA, TS, and MJE contributed to the GEOS-Chem halogen model development. BHL, JDH, FDL, and JAT conducted and processed the measurement during WINTER campaign. GLH conducted and processed the measurement during SEAC4RS and KORUS-AQ campaigns. XW and DJJ prepared the manuscript with contributions from all co-authors.

**Data availability.** The model code is available from the corresponding author upon request, and will be made available to the community through the standard GEOS-Chem (http://www.geos-chem.org) in the future. Data of WINTER campaign are available to the general public at https://www.eol.ucar.edu/field_projects/winter. Data of NASA SEAC4RS AND KORUS-AQ missions and are available to the general public through the NASA data archive (https://www-air.larc.nasa.gov/cgi-bin/ArcView/seac4rs and https://www-air.larc.nasa.gov/cgi-bin/ArcView/korusaq). IMPROVE data are available through the Federal Land Manager Environmental database (http://views.cira.colostate.edu/fed).

**Acknowledgments.** This work was supported by the Atmospheric Chemistry Program of the US National Science Foundation and by the Joint Laboratory for Air Quality and Climate (JLAQC) between Harvard and the Nanjing University for Information Science and Technology (NUIST). QC and BA were supported by National Science Foundation (AGS 1343077). We thank Prasad S. Kasibhatla for the insightful discussion. WINTER data are provided by NCAR/EOL under sponsorship of the National Science Foundation (https://www.eol.ucar.edu/field_projects/winter). SEAC4RS and KORUS-AQ data are provided by NASA LaRC Airborne Science Data for Atmospheric Composition (https://www-air.larc.nasa.gov). IMPROVE is a collaborative association of state, tribal, and federal agencies, and international partners. US Environmental Protection Agency is

1    the primary funding source, with contracting and research support from the National Park Service. The Air Quality

2    Group at the University of California, Davis is the central analytical laboratory, with ion analysis provided by Research

3    Triangle Institute, and carbon analysis provided by Desert Research Institute.

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

**Table 1: Global sources and sinks of gas-phase inorganic (Cl$_y$) and reactive (Cl*) tropospheric chlorine [a].**

| | Cl$_y$ (Gg Cl a$^{-1}$) | Cl* (Gg Cl a$^{-1}$) |
|---|---|---|
| Total source | 75200 | 25000 |
| Sea Salt | 63900 | 11900 |
| Acid displacement [b] | 52000 | - |
| HOBr + Cl$^-$ | 8590 | 8590 |
| N$_2$O$_5$ + Cl$^-$ | 1810 | 1810 |
| HOI, IONO$_x$ [c] + Cl$^-$ | 641 | 641 |
| ClNO$_2$ + Cl$^-$ | 327 | 327 |
| OH + Cl$^-$ | 403 | 403 |
| ClNO$_3$ + Cl$^-$ | 64 | 64 |
| HOCl + Cl$^-$ | 61 | 61 |
| HCl + OH | - | 9720 |
| Organochlorines | 3320 | 3300 |
| CH$_3$Cl + OH [d] | 2200 | 2180 |
| CH$_2$Cl$_2$ + OH | 780 | 780 |
| CHCl$_3$ + OH | 298 | 298 |
| CH$_2$ICl + OH | 46 | 46 |
| Stratosphere [e] | 380 | 64 |
| Anthropogenic HCl [f] | (6660) | - |
| Open fires | 7640 | - |
| Total sink | 75200 | 25000 |
| Deposition | 71400 | 346 |
| Dry | 35200 | 170 |
| Wet | 36200 | 176 |
| Uptake by alkaline SSA | 3800 | - |
| Conversion to HCl [g] | - | 24600 |
| Tropospheric mass (Gg) | 316 | 12 |
| Lifetime (hours) | 37 | 3.8 |

[a] Annual totals for 2016 computed from GEOS-Chem. Gas-phase inorganic chlorine is defined as Cl$_y$ ≡ Cl + 2×Cl$_2$ + 2×Cl$_2$O$_2$ +
ClNO$_2$ + ClNO$_3$ + ClO + ClOO + OClO + BrCl + ICl + HOCl + HCl. Reactive chlorine is defined as Cl* ≡ Cl$_y$ – HCl. Thus the
source of HCl can be inferred from the Table entries as Cl$_y$ – Cl*. The definition of Cl$_y$ excludes aerosol Cl$^-$, which has a very large
sea salt source of 1780 Tg Cl a$^{-1}$ but is mainly removed by deposition, HCl is the dominant component of Cl$_y$ but is also mostly
removed by deposition. Reactive chlorine Cl* is the chemical family principally involved in radical cycling.
[c] IONO$_x$ ≡ IONO + IONO$_2$
[b] Net production minus loss of HCl from acid aerosol displacement by HNO$_3$ and H$_2$SO$_4$ computed as thermodynamic equilibrium.
[d] The source from CH$_3$Cl + Cl is not shown since it contributes < 1% of CH$_3$Cl oxidation. Same for other organochlorines.
[e] Net stratospheric input to the troposphere.
[f] Coal combustion, waste incineration, and industrial activities. These emissions are only included in a sensitivity simulation (see
Section 2.2 and 4.2 for details) and are therefore listed here in parentheses. Emissions of anthropogenic fugitive dust are estimated
as less than 390 Gg a$^{-1}$ (Section 2,2) and are not included in the model.
[g] From reactions of Cl atoms (see Figure 1).

1    **Table 2: Heterogeneous reactions of Cl⁻and reactive uptake coefficients (γ) [a].**

| | Reaction | Reactive uptake coefficient ($\gamma$) | Footnote |
|---|---|---|---|
| R3 | $N_2O_5 + \varphi Cl^- + (1-\varphi)H_2O \rightarrow \varphi ClNO_2 + (2-\varphi)NO_3^- + 2(1-\varphi)H^+$ | $\gamma = Bk'_{2f}\left(1 - \dfrac{1}{\left(\frac{k_3[H_2O]}{k_{2b}[NO_3^-]}\right) + 1 + \left(\frac{k_4[Cl^-]}{k_{2b}[NO_3^-]}\right)}\right)$ $k'_{2f} = \beta(1 - e^{-\delta[H_2O]})$ ; $\varphi = (\frac{k_2[H_2O]}{k_3[Cl^-]} + 1)^{-1}$ $B = 3.2 \times 10^{-8}s$ ; $k_3/k_2 = 450$ $\beta = 1.15 \times 10^6 s^{-1}$ ; $\delta = 0.13M^{-1}$ $k_3/k_{2b} = 0.06$ ; $k_4/k_{2b} = 29$ | b |
| R4 | $OH + Cl^- \rightarrow 0.5Cl_2 + OH^-$ | $\gamma = 0.04[Cl^-]$ | c |
| R5 | $HOBr + Cl^- + H^+ \rightarrow BrCl + H_2O$ | $\gamma = \left(\frac{1}{\Gamma_b} + \frac{1}{\alpha_b}\right)^{-1}$ $\Gamma_b = 4H_{HOBr}RTI_r k_b[Cl^-][H^+]f(r,I_r)/c$ $I_r = \sqrt{D_l/(k_b[Cl^-][H^+])}$ ; $\alpha_b = 0.6$ $k_b = 2.3 \times 10^{10}M^{-2}s^{-1}$; $D_l = 1.4 \times 10^{-5}cm^2s^{-1}$ | d |
| R6 | $ClNO_3 + Cl^- \rightarrow Cl_2 + NO_3^-$ | $\gamma = 0.0244$ | e |
| R7 | $ClNO_2 + Cl^- \rightarrow NO_2^- + Cl_2$ | $\gamma = \left(\frac{1}{\Gamma_b} + \frac{1}{\alpha_b}\right)^{-1}$ (pH < 2), $\gamma = 0$ (pH > 2) $\Gamma_b = 4H_{ClNO_2}RTI_r k^{II}[Cl^-]f(r,I_r)/c$ $I_r = \sqrt{D_l/(k^{II}[Cl^-])}$ ; $\alpha_b = 0.01$ $k^{II} = 10^7M^{-2}s^{-1}$ ; $D_l = 1 \times 10^{-5}cm^2s^{-1}$ | |
| R8 | $ClNO_2 + Br^- \rightarrow NO_2^- + BrCl$ | $\gamma = \left(\frac{1}{\Gamma_b} + \frac{1}{\alpha_b}\right)^{-1}$ $\Gamma_b = 4H_{ClNO_2}RTI_r k^{II}[Br^-]f(r,I_r)/c$ $I_r = \sqrt{D_l/(k^{II}[Br^-])}$ ; $\alpha_b = 0.01$ $H_{ClNO_2}{}^2 D_l k^{II} = 0.101Mcm^2s^{-2}$ | |
| R9 | $HOCl + Cl^- + H^+ \rightarrow Cl_2 + H_2O$ | $\gamma = min(\left(\frac{1}{\Gamma_b} + \frac{1}{\alpha_b}\right)^{-1}, 2 \times 10^{-4})$ $\Gamma_b = 4H_{HOCl}RTI_r k_t[Cl^-][H^+]f(r,I_r)/c$ $I_r = \sqrt{D_l/(k_t[Cl^-][H^+])}$ ; $\alpha_b = 0.8$ $k_t = 1.5 \times 10^4M^{-2}s^{-1}$ ; $D_l = 2 \times 10^{-5}cm^2s^{-1}$ | |
| R10 | $NO_3 + Cl^- \rightarrow NO_3^- + Cl^-$ | $\gamma = \left(\frac{1}{\Gamma_b} + \frac{1}{\alpha_b}\right)^{-1}$ $\Gamma_b = 4H_{NO_3}RTI_r k'[Cl^-]f(r,I_r)/c$ $I_r = \sqrt{D_l/(k'[Cl^-])}$ ; $\alpha_b = 0.013$ $k' = 2.76 \times 10^6M^{-2}s^{-1}$ ; $D_l = 1 \times 10^{-5}cm^2s^{-1}$ | |
| R11 | $IONO_2 + Cl^- \rightarrow ICl + NO_3^-$ | $\gamma = 8.5 \times 10^{-3}$ | f |
| R12 | $IONO + Cl^- \rightarrow ICl + NO_2^-$ | $\gamma = 0.017$ | f |
| R13 | $HOI + Cl^- \rightarrow ICl + OH^-$ | $\gamma = 8.5 \times 10^{-3}$ | f |

[a] Formulations for the reactive uptake coefficient γ are from IUPAC (Ammann et al., 2013) unless stated otherwise in the footnote
column. Brackets denote aqueous-phase concentrations in unit of M (moles per liter of water). $R$ is the ideal gas constant. $c$ is the
average gas-phase thermal velocity for the reactant with $Cl^-$. The reactive uptake coefficient is used to calculate the reaction rate
following equation (1). $f(r, I_r) = \coth(r/I_r) - (I_r/r)$ is a spherical correction to mass transfer where $I_r$ is a reacto-diffusive length
scale and $r$ is the radius of the aerosol particle or cloud droplet.
[b] Bertram and Thornton (2009); Roberts et al. (2009).
[c] Knipping and Dabdub (2002)
[d] $k_b$ is based on Liu and Margerum (2001). R5 competes with the heterogeneous reactions $HOBr+Br^-$ and $HOBr+S(IV)$ as given by
Chen et al. (2017). The BrCl product may either volatilize or react with $Br^-$ to produce $Br_2$ and return $Cl^-$ following Fickert et al.
(1999), as  described in Section 5.2.
[e] Assumes that $Cl^-$ is present in excess so that γ does not depend on $[Cl^-]$. However, R6 competes with the heterogeneous
reaction $ClNO_3+Br^-$ as given by Schmidt et al. (2016), with the branching ratio determined by the relative rates.
[f] These reactions are based on Sherwen et al. (2016a) and only take place in SSA.
**Table 3: Chloride deficits in sea salt aerosol.[a]**

| Location | Modeled Cl⁻ deficit (%) | Measured Cl⁻ deficit (%) |
|---|---|---|
| North Carolina coast | +40 | -1 to +90 |
| Townsville coast, Australia | +23 | +33 |
| California coast | +21 | +2 to +75 |
| Greenland Sea | +18 | +6 to +22 |
| North Atlantic Ocean | +14 | -24 to +54 |
| Equatorial Atlantic | +12 | +11 to +64 |
| Puerto Rico coast | +9 | +7 to +25 |
| Pacific Ocean | +6 | -22 ~ +40 |
| Cape Grim, Australia | +4 | -50~+15 |

[a] Deficits relative to seawater composition. Observations compiled by Graedel and Keene (1995) are reported there as ranges for
individual regions and months, with the ranges likely reflecting measurement uncertainty rather than physical variability.  Model
values are means for the regions and months of observations.
**Table 4: Surface air mixing ratios of reactive chlorine (Cl*)[a]**

| Location | Modeled Cl* (ppt) | Measured mean Cl* (ppt) | Reference |
|---|---|---|---|
| Atlantic cruise near Europe | 43 | 27[b] | Keene et al. (2009) |
| Appledore Island (US east coast) | 17 | < 20 | Keene et al. (2007) |
| Atlantic cruise near North Africa | 5 | < 24 | Keene et al. (2009) |
| Southern Ocean cruise | 4 | < 24 | Keene et al. (2009) |
| Hawaii | 4 | 6 | Pszenny et al. (2004) |
| Tropical Atlantic cruise | 2 | < 24 | Keene et al. (2009) |
| Alert (Canada) | 0.2 | < 14 | Impey et al. (1999) |

[a] Reactive chlorine Cl* is the ensemble of gas-phase inorganic chlorine species excluding HCl. Measurements are 24- hour
averages. Model values are monthly means in 2016 taken for the same month and location as the observations.
[b] Median value

**Table 5: Comparison of modeled maximum ClNO₂ mixing ratios to surface observations [a]**

| Location | Date | Observed (ppt) | Simulated (ppt) | References |
|---|---|---|---|---|
| Manchester, UK | Oct-Nov 2014 | 510 | 400 | Priestley et al. (2018) |
| Weybourne, UK | Jun-Aug 2015 | 1100 | 1200 | Sommariva et al. (2018) |
| East Anglia Coast, UK | Jan 2014 | 100 | 400 | Bannan et al. (2017) |
| Leicester, UK | Feb 2016 | 730 | 760 | Sommariva et al. (2018) |
| London, UK | July-Aug 2012 | 730 | 510 | Bannan et al. (2015) |
| Calgary, Canada | Apr 2010 | 240 | 130 | Mielke et al. (2011) |
| Calgary, Canada | Sep 2010-Mar 2011 | 340 | 170 | Mielke et al. (2015) |
| Penlee Point, UK | Apr-May 2015 | 920 | 870 | Sommariva et al. (2018) |
| Kleiner Feldberg, Germany | Aug-Sep 2011 | 850 | 400 | Phillips et al. (2012) |
| Long Island Sound | Mar 2008 | 200 | 210 | Kercher et al. (2009) |
| Olympic Park, South Korea | May-Jun 2016 | 780 | 520 | Jeong et al. (2018) |
| Taehwa Research Forest, South Korea | May-Jun 2016 | 2600 | 220 | Jeong et al. (2018) |
| Boulder, Colorado | Feb 2009 | 440 | 130 | Thornton et al. (2010) |
| Pasadena, California | May-Jun 2010 | 3500 | 360 | Mielke et al. (2013) |
| Offshore of Los Angeles, California | May-Jun 2010 | 1800 | 500 | Riedel et al. (2013) |
| La Jolla, California | Feb 2013 | 65 | 65 | Kim et al. (2014) |
| Houston, Texas | Aug-Sep 2006 | 1200 | 150 | Osthoff et al. (2008) |
| Houston, Texas | Sep 2013 | 140 | 18 | Faxon et al. (2015) |
| Hong Kong, China | Aug, 2012 | 1900 | 200 | Tham et al. (2014) |
| Hong Kong, China | Nov-Dec, 2013 | 4700 | 410 | Wang et al. (2016) |

[a] Observed and modeled values are maxima for the reporting period. Model maxima are based on hourly values sampled at the same location and time period as the observations. The sites are listed in order of decreasing latitude.

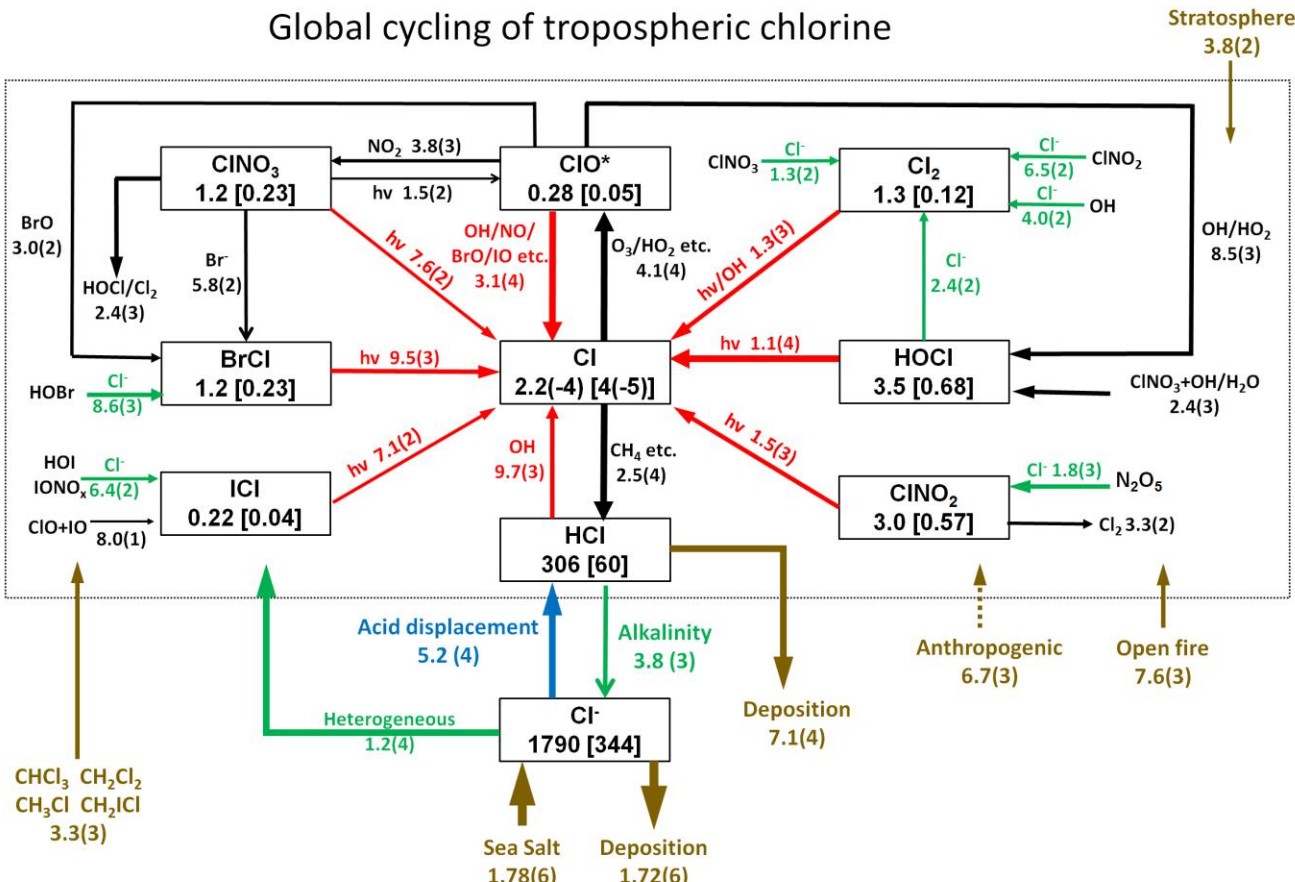

**Figure 1: Global budget and cycling of tropospheric inorganic chlorine (Cl$_y$) in GEOS-Chem. The figure shows global**
**annual mean rates (Gg Cl a$^{-1}$), masses (Gg), and mixing ratios (ppt, in brackets) for simulation year 2016. Read 2.5(4) as**
**2.5×10$^4$. ClO\* stands for ClO + OClO + ClO$_2$ + 2×Cl$_2$O$_2$; 84% is present as ClO. Reactions producing Cl atoms and related**
**to Cl$^-$ heterogeneous chemistry are shown in red and green, respectively. The dotted box indicates the Cl$_y$ family, and arrows**
**into and out of that box represent general sources and sinks of Cl$_y$. Reactions with rate < 100 Gg Cl a$^{-1}$ are not shown.**
**Anthropogenic emissions of HCl as indicated by a dashed line are only included in a sensitivity simulation (see Section 2.2**
**and 4.2 for details).**

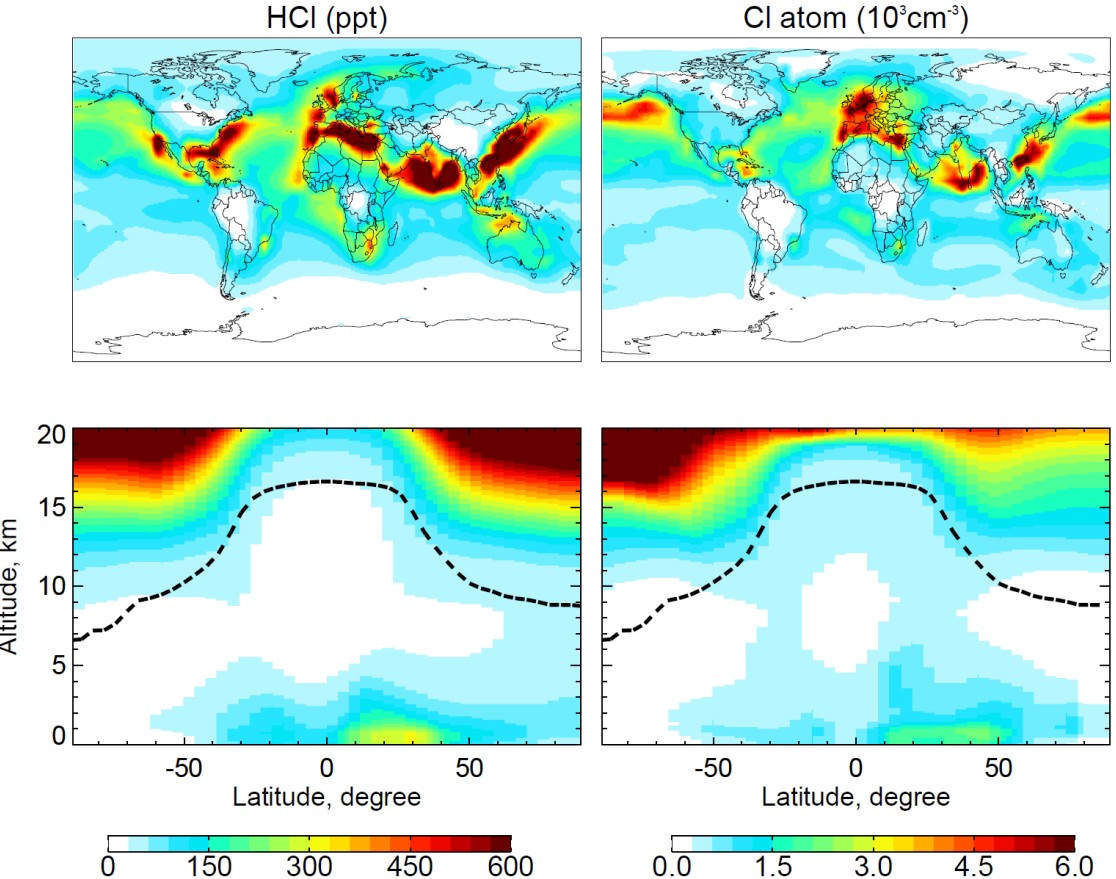

**Figure 2: Global distributions of annual mean HCl mixing ratios and Cl atom concentrations in GEOS-Chem. The top panels show surface air mixing ratios/concentrations. The bottom panels show zonal mean mixing ratios/concentrations as a function of latitude and altitude. Dashed lines indicate the tropopause.**

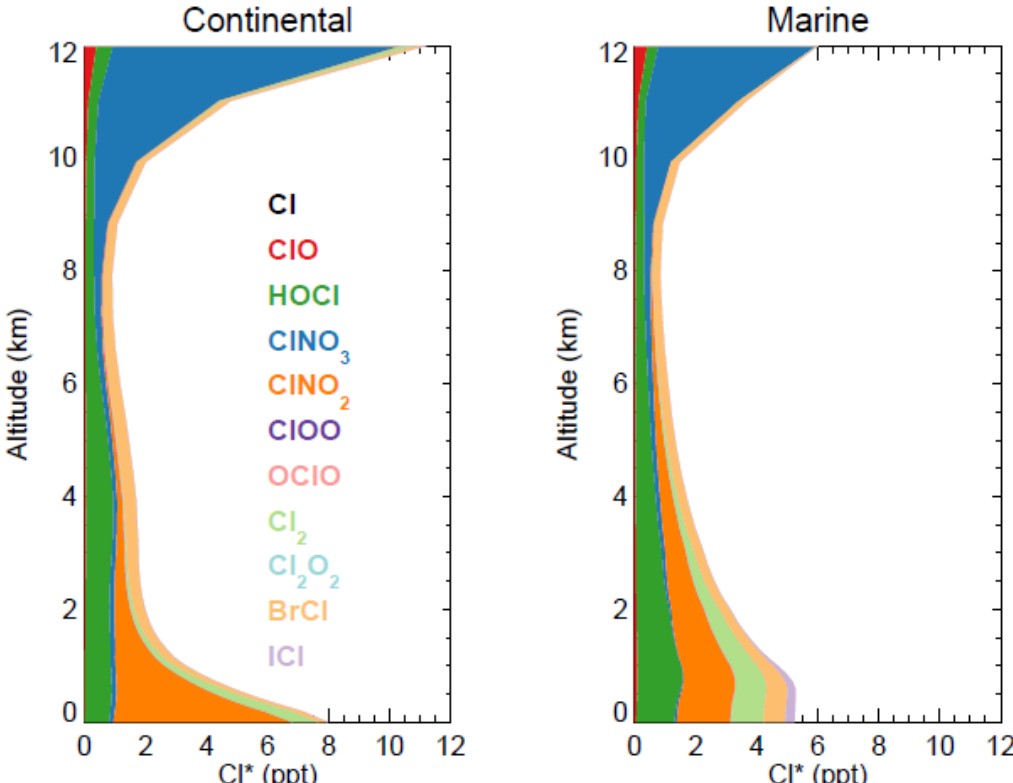

Figure 3: Global annual mean vertical distributions of reactive chlorine species (Cl*) in GEOS-Chem for continental and marine air. Stratospheric conditions are excluded.

**Figure 4: HCl and HNO₃ surface air mixing ratios at coastal/island sites and from ocean cruises, arranged from left to right in order of increasing latitude. Observations are means (black circles) or medians (black triangles) depending on availability. Model values are monthly means for the sampling locations. Also shown are results from a sensitivity simulation with no mobilization of Cl⁻ from sea salt aerosol (SSA). References: (1, 2, 5, 10) Keene et al. (2009); (3) Sanhueza and Garaboto (2002); (4) Sander et al. (2013); (6) Dasgupta et al. (2007); (7) Crisp et al. (2014); (8) Bari et al. (2003); (9) Keene et al. (2007).**

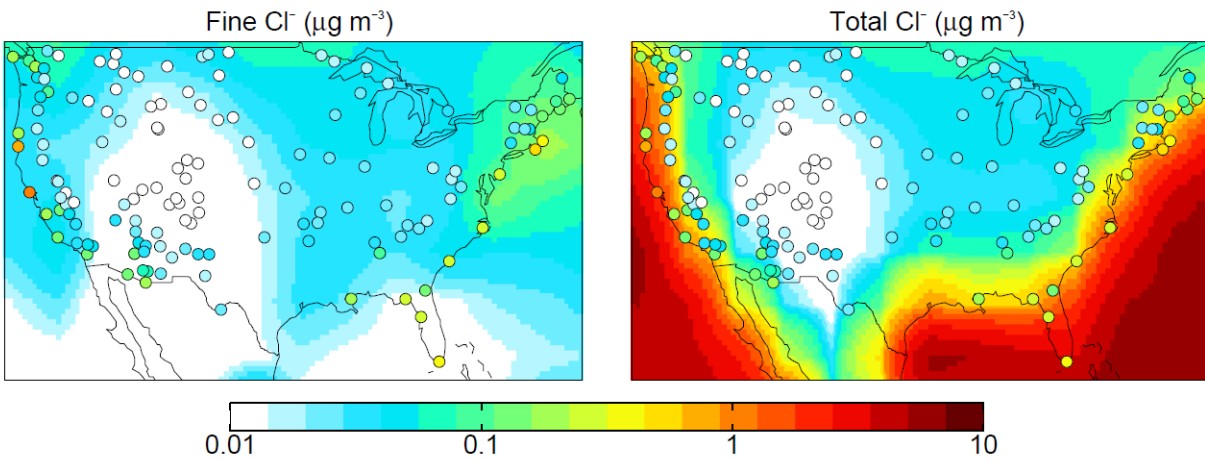

**Figure 5: Aerosol Cl⁻ concentrations in surface air over the contiguous US. Values are annual means for 2016. GEOS-Chem**
**model values are shown as contours separately for fine Cl⁻ (<1 μm diameter) and total Cl⁻. Observations from the IMPROVE**
**network (<2.5 μm diameter) are shown as circles and are the same in both panels; one would expect them to be higher than**
**the model fine Cl⁻ but lower than total Cl⁻.**

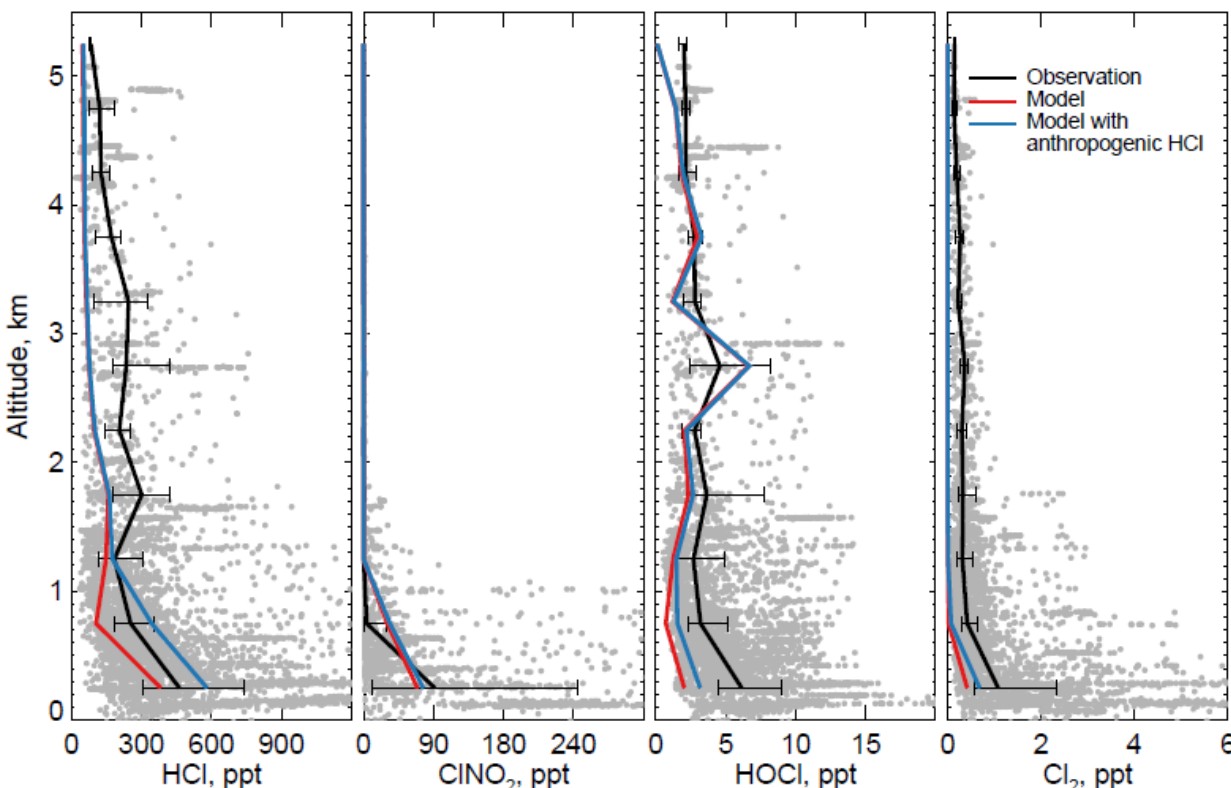

**Figure 6: Vertical profiles of HCl, nighttime ClNO₂, HOCl, and Cl₂ mixing ratios during the WINTER campaign over the eastern US and offshore in February-March 2015. Observations from Haskins et al. (2018) are shown as individual 1-minute data points, with medians and 25th-75th percentiles in 500-m vertical bins. Measurements below the detection limit are treated as the median of 0 and detection limit. ClNO₂ data exclude daytime (10:00-16:00 local) when mixing ratios are near zero both in the observations and in the model (Figure 7). Model values are shown as medians sampled along the flight tracks. Also shown are results from a sensitivity simulation including the anthropogenic chlorine inventory of McCulloch et al. (1999).**

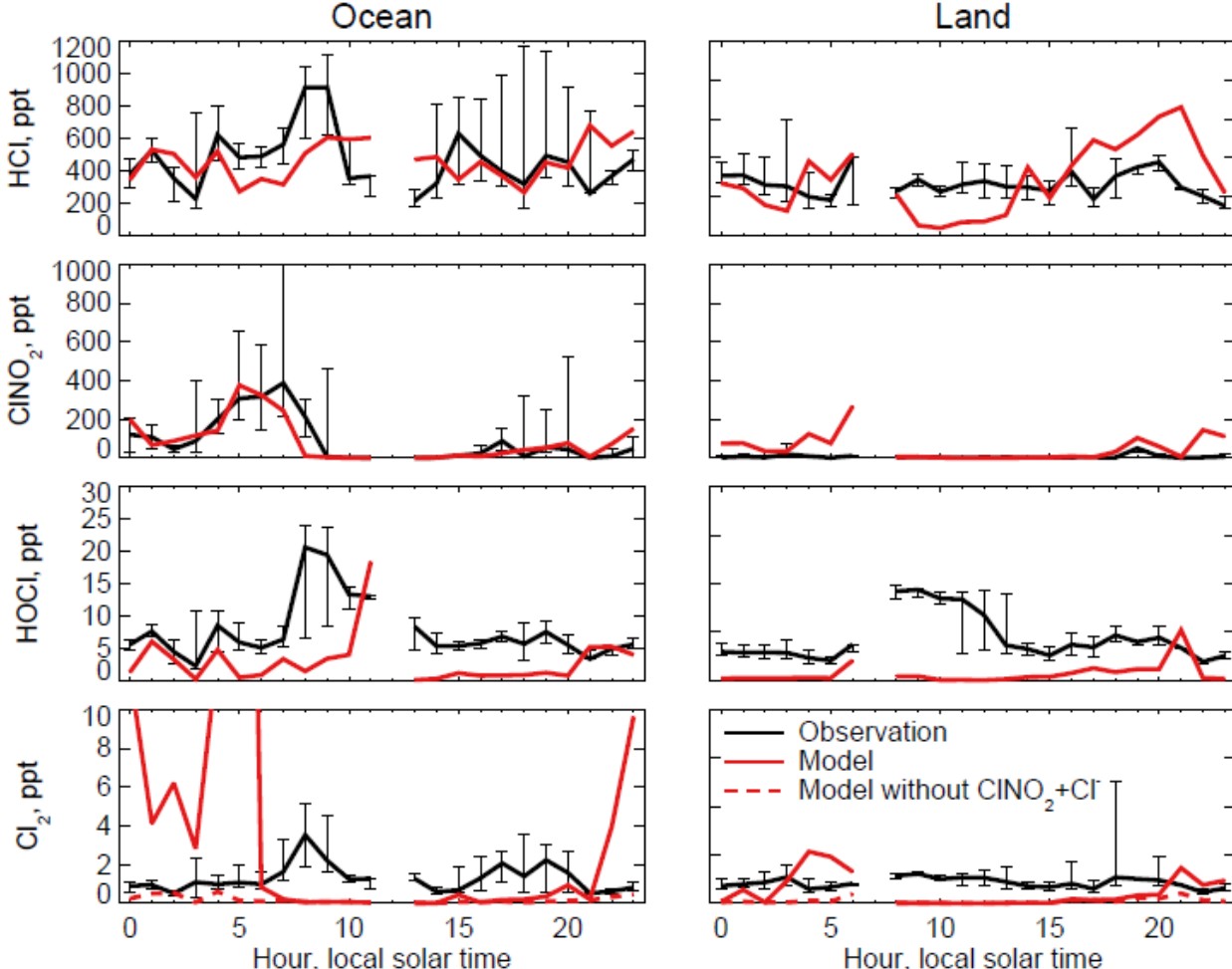

Figure 7: Median diurnal variations of HCl, ClNO₂, and Cl₂ mixing ratios below 1 km altitude during the WINTER aircraft campaign over the eastern US and offshore in February-March 2015. The data are separated between ocean (left panels) and land (right panels). Model values are compared to observations from Haskins et al. (2018). Vertical bars show the 25th-75th percentiles in the observations. Measurements below the detection limit are treated as the median of 0 and detection limit. Also shown are results from a sensitivity simulation excluding ClNO₂+Cl⁻, which has negligible effect on HCl, ClNO₂, and HOCl, but brings the Cl₂ simulation in much better agreement with observations at night.

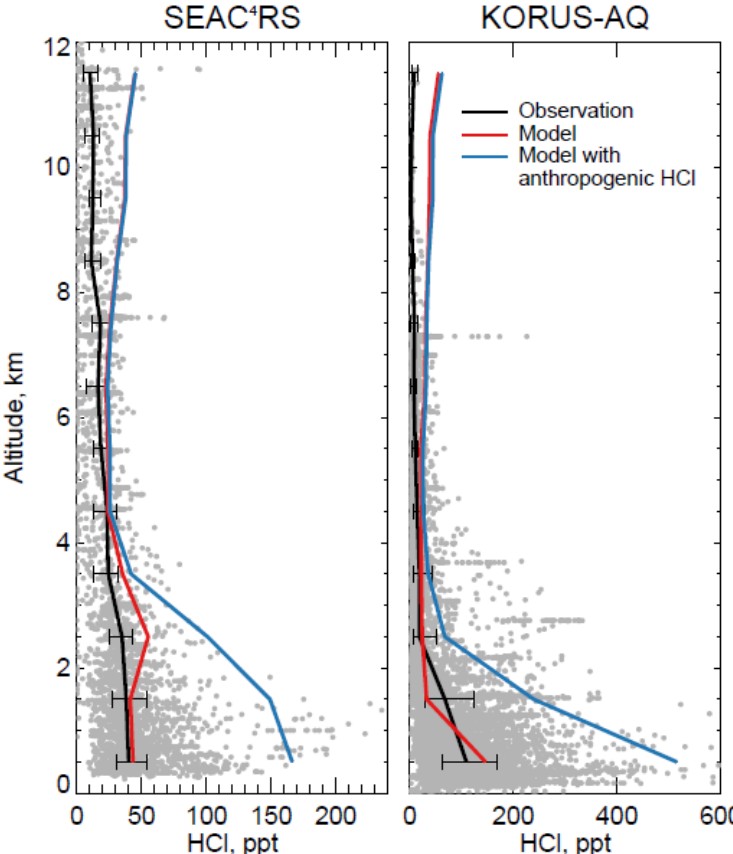

**Figure 8: Vertical profiles of HCl mixing ratios during the SEAC[4]RS aircraft campaign over the Southeast US (95º-81.5°W, 30.5º-39°N) in August-September 2013 and during the KORUS-AQ aircraft campaign over and around the Korean peninsula (120º-132°E, 32°-38°N) in May-June 2015. Observations from the Georgia Tech CIMS instrument are shown as gray points (1-minute averages), with medians and 25[th]-75[th] percentiles in 1-km vertical bins. Model values are sampled along the flight tracks and for the measurement period. Measurements below the detection limit are treated as the median of 0 and detection limit.**

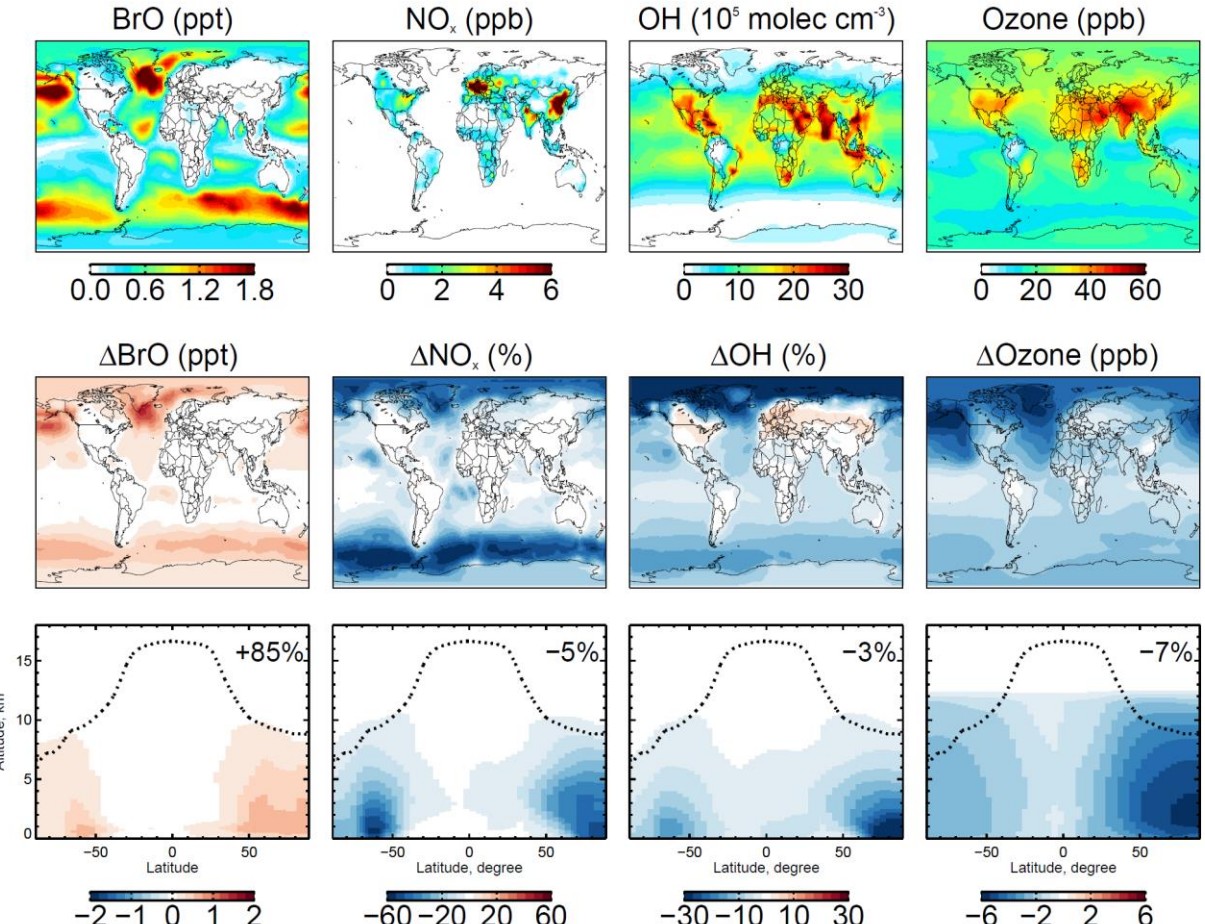

**Figure 9: Effects of tropospheric chlorine chemistry on BrO, NOₓ, OH, and ozone. The top panels show the annual mean surface concentrations of BrO, NOₓ, OH, and ozone simulated in our standard model including tropospheric chlorine chemistry. The lower panels show the changes in annual mean mixing ratios/concentrations due to tropospheric chlorine chemistry, as determined by difference with a sensitivity simulation including no Cl$_y$ production and cycling. The middle panels show the changes in surface air concentrations and the bottom panels show the changes in zonal mean mixing concentrations as a function of latitude and altitude. Black dashed lines indicate the tropopause. Numbers in bottom panels show the global tropospheric mean differences.**

