# Peer review of "The role of chlorine in global tropospheric chemistry"

_Atmospheric Chemistry and Physics, 2018_

## Referee Comment (RC1) · Anonymous Referee #1 · 26 Nov 2018

This paper presents results describing the application of the GEOS-CHEM model to understanding the role of chlorine chemistry in the troposphere. The paper makes some new contributions and improves upon previous GEOS-CHEM modeling of halogen chemistry, but has so many major flaws and omissions that will require considerable modification. Since these changes may impact some of the findings, it is hard to know how to assess its publishability at this point. I have the following general and specific comments and questions.

General Comments:

There are number of errors and omissions in the reference list: papers that are noted in the text, but not in list, listed with errors, or just the wrong paper. Here is a list roughly in order of appearance: Liu et al., 2001 – which one, there are two in the reference list? Jaegle et al., 2012 – do you mean Jaegle 2011 here? Wesely 1989 is not in the

list. Wang 1998 is not in the list. Jaegle et al., 2010 – do you mean Jaegle 2011 here? Martin et al., 2002 – do you mean 2003? Abbatt and Wachewsky, 1998 is not in the list Bey et al., 2001 is not in the list. Pszenny et al., 1991 – do you mean 1993 here? Roberts et al., 2008 is not in the list. Allen et al., 2007 is not in the list unless you mean Allan et al. 2011 Wang et al., 1994 is not in the list. Roberts et al. 2009 is not the correct reference, it should be this Roberts et al., 2009 (Roberts et al., 2009) Mielke et al., 2013 is not on the list.

Chloride Sources: The paper ignores soil and wind-blown dust sources of particle chloride. Those were described by Sarwar et al., 2012 (Sarwar et al., 2012) (which by the way is not referenced in this paper) as the major sources of chloride in the Eastern U.S. and therefore will drive the chlorine budget in the middle of continents. This paper needs to give serious consideration to this source, and can use the IMPROVE network chloride data as a means to assess this continental source.

Comparisons:

The selection of data sets for comparison was certainly not thorough, and obvious opportunities for comparison were apparently not considered.

The paper seems to lean heavily on the Graedel and Keene work from 1995, but I believe further work has been done since then. There are comparisons that could be made that would greatly add to the model analysis and there are a number of data sets that were not included in the comparisons that were done.

Particle chloride can be compared to IMPROVE data (see for example Sarwar et al., 2012) to see how the model does. This is a long-term data set of considerable geographic extent.

There are observations of HCl displacement by HNO3 in the LA Basin that need to be considered for comparison (Gard et al., 1998).

There are a number of HCl data sets that were not compared with. Here is at least a

partial list: Kim et al., INTEX-B (Kim et al., 2008), which includes one of the co-authors. CalNex ground site data for both particle chloride and gas phase HCl can be found at: https://www.esrl.noaa.gov/csd/projects/calnex/.

There are a considerable number of ClNO2 data sets that were not compared with: Osthoff et al., 2009 – this covers essentially urban/industrial Houston and the near coastal environment. Sommariva et al., 2018 (Sommariva et al., 2018) covers several sites in England, including coastal sites. Le Breton et al., 2018 (Le Breton et al., 2018) Priestley et al., 2018. (Priestley et al., 2018) Tham et al., 2016 (Tham et al., 2013) Wang , Z. et al., (Wang et al., 2017b) Wang X., et al., 2017 (Wang et al., 2017a) Riedel et al, 2013 (Riedel et al., 2013) Mielke et al., 2015 (Mielke et al., 2015) Bannan et al., 2017 (Bannan et al., 2017) Young et al., 2012 (Young et al., 2012)(data set can be found at https://www.esrl.noaa.gov/csd/projects/calnex/) Zhou, et al., 2018 (Zhou, 2018)

It's true that there is only one lab study of the ClNO2 + Cl- Chemistry (Roberts et al., 2008), but the authors seem to have missed a key point of that paper: that reaction seems to only occur at pH=2 and below. Unless I missed something, this paper seems to allow this reaction after the alkalinity of the aerosol is neutralized. This may account for why the Cl2 is to high compared to the WINTER observations.

Specific Comments:

Page 10, Line 10, and Table 5. There are several data sets for which the model is a factor of 10 low, so it is hard to say the model performed "credibly". Also, most if not all of these data sets are available, so could be compared more in depth, not just maxima.

Page 11, lines 2&3. I believe the aircraft did missed approaches over airfields that could be used to do near-surface comparisons.

Page 14, Line 22. Should be 'troposphere'.

References:

[Figure]

Bannan, T. J., Bacak, A., Le Breton, M., Flynn, M., Ouyang, B., McLeod, M., Jones, R., Malkin, T. L., Whalley, L. K., Heard, D. E., Bandy, B., Khan, M. A. H., Shallcross, D. E., and Percival, C. J.: Ground and airborne U.K. measurements of nitryl chloride: An investigation of the role of Cl atom oxidation at Weybourne Atmospheric Observatory, J. Geophys. Res.-Atmos., 122, 11,154-111,165, 2017.

Gard, E. E., Kleeman, M. J., Gross, D. S., Hughes, L. S., Allen, J. O., Morrical, B. D., Fergenson, D. P., Dienes, T., Galli, M. E., Johnson, R. J., Cass, G. R., and Prather, K. A.: Direct observation of heterogeneous chemistry in the atmosphere, Science, 279, 1184-1187, 1998.

Kim, S., Huey, L. G., Stickel, R. E., Pierce, R. B., Chen, G., Avery, M. A., Dibb, J. E., Diskin, G. S., Sachse, G. W., McNaughton, C. S., Clarke, A. D., Anderson, B. E., and Blake, D. R.: Airborne measurements of HCl from the marine boundary layer to the lower stratosphere over the North Pacific Ocean during INTEX-B, Atmos. Chem. Phys. Discuss., 8, 3563-3595, 2008.

Le Breton, M., Hallquist, Å. M., Pathak, R. K., Simpson, D., Wang, Y., Johansson, J., Zheng, J., Yang, Y., Shang, D., Wang, H., Liu, Q., Chan, C., Wang, T., Bannan, T. J., Priestley, M., Percival, C. J., Shallcross, D. E., Lu, K., Guo, S., Hu, M., and Hallquist, M.: Chlorine oxidation of VOCs at a semi-rural site in Beijing: significant chlorine liberation from ClNO2 and subsequent gas- and particle-phase Cl–VOC production, Atmos. Chem. Phys., 18, 13013-13030., 2018.

Mielke, L. H., Furgeson, A., O'dame- Ankrah, C. A., and Osthoff, H. D.: Ubiquity of ClNO2 inthe urban boundary layer of Calgary, AB, Canada. , Canadian J. Chem., 94, 414-423, 2015.

Priestley, M., le Breton, M., Bannan, T. J., Worrall, S. D., Bacak, A., Smedley, A. R. D., Reyes-Villegas, E., Mehra, A., Allan, J., Webb, A. R., Shallcross, D. E., Coe, H., and Percival, C. J.: Observations of organic and inorganic chlorinated compounds and their contribution to chlorine radical concentrations in an urban environment in northern

Europe during the wintertime, Atmos. Chem. Phys., 18, 13481-13493, 2018.

Riedel, T. P., Wagner, N. L., Dube, W. P., Middlebrook, A. M., Brock, C. A., Young, C. J., Ozturk, F., Bahreini, R., VandenBoer, T. C., Wolfe, D., Williams, E. J., Roberts, J. M., Brown, S. S., and Thornton, J. A.: Chlorine activation within urban or power plant plumes: Vertically resolved ClNO2 and Cl2 measurements from a tall tower in a polluted continental setting, J. Geophys. Res., 118, 2013.

Roberts, J. M., Osthoff, H. D., Brown, S. S., and Ravishankara, A. R.: N2O5 Oxidizes Chloride to Cl2 in Acidic Atmospheric Aerosol, Science, 321, 1059., 2008.

Roberts, J. M., Osthoff, H. D., Brown, S. S., Ravishankara, A. R., Coffman, D., Quinn, P. K., and Bates, T. S.: Laboratory studies of products of N2O5 Uptake on Cl- Containing Substrates, Geophys. Res. Lett., 36, L20808, 2009.

Sarwar, G., Simon, H., Bhave, P., and Yarwood, G.: Examining the impact of heterogeneous nitryl chloride production on air quality across the United States., Atmos. Chem. Phys., 12, 6455-6473, 2012.

Sommariva, R., Hollis, L. D. J., Sherwen, T., Baker, A. R., Ball, S. M., Bandy, B. J., Bell, T. G., Chowdhury, M. N., Cordell, R. L., Evans, M. J., Lee, J. D., Reed, C., Reeves, C. E., Roberts, J. M., Yang, M., and Monks, P. S.: Seasonal and geographical variability of nitryl chloride and its precursors in Northern Europe, Atmospheric Science Letters, 19, 1-10, 2018.

Tham, Y. J., Yan, C., Xue, L., Zha, Q., Wang, X. B., and Wang, T.: Presence of high nitryl Chloride in Asian coastal environment and its impact on atmospheric photochemistry, Chin. Sci. Bull., doi: 10.1007/s11434-013-0063-y, 2013. 2013.

Wang, X., Wang, H., Xue, L., Wang, T., Wang, L., Gu, R., Wang, W., Than, Y. T., Wang, Z., Yang, I., Chen, J., and Wang, W.: Observations of N2O5 and ClNO2 at a polluted urban surface site in North China: High N2O5 uptake coefficients and low ClNO2 product yields„ Atmos Environ., 156, 125-134, 2017a.

[Figure]

Wang, Z., Wang, W., Tham, Y. J., Li, Q., Wang, H., Wen, L., Wang, X., and Wang, T.: Fast heterogeneous N2O5 uptake and ClNO2 production in power plant and industrial plumes observed in the nocturnal residual layer over the North China Plain, Atmos. Chem. Phys., 17, 12361-12378, 2017b.

Young, C. J., Washenfelder, R. A., Roberts, J. M., Mielke, L. H., Osthoff, H. D., Tsai, C., Pikelnaya, O., Stutz, J., Veres, P. R., Cochran, A. K., VandenBoer, T. C., Flynn, J., Grossberg, N., Haman, C. L., Lefer, B., Stark, H., Graus, M., de Gouw, J., Gilman, J. B., Kuster, W. C., and Brown, S. S.: Vertically resolved measurements of nighttime radical reservoirs in Los Angeles and their contribution to the urban radical budget, Environ. Sci. Technol., 46, 10965-10973, 2012.

Zhou, W., Zhao, J., Ouyang, B., Mehra, A., Xu, W., Wang, Y., Bannan, T. J., Worrall, S. D., Priestley, M., Bacak, A., Chen, Q., Xie, C., Wang, Q., Wang, J., Du, W., Zhang, Y., Ge, X., Ye, P., Lee, J. D., Fu, P., Wang, Z., Worsnop, D., Jones, R., Percival, C. J., Coe, H., and Sun, Y.: Production of N2O5 and ClNO2 in summer in urban Beijing, China, Atmos. Chem. Phys., 18, 11581-11597, 2018.

---

## Referee Comment (RC2) · Anonymous Referee #2 · 27 Nov 2018

This paper by Wang et al. presents an extensive modelling study of chlorine chemistry using GEOS-Chem. The updates to the GEOS-Chem chemical mechanism are much needed. I think the paper is suitable for ACP, after the authors have addressed the following points.

MAJOR COMMENTS

As a general comment, I think it would be useful to have a more detailed discussion of the difference in the results between the version of GEOS-Chem with the previous mechanism and this new version. As it stands, the reader is mostly referred to previous papers. Figure 9 provides some of this information but it is presented as a sensitivity study, so it is not really a comparison of the old and new mechanisms.

Another general point is that the choice of datasets to compare with the model seems somewhat arbitrary. I understand it is not possibile to use all the available datasets, but

you have to explain why you selected certain datasets and not others for this work.

On page 8: you say that the BrCl concentration in this work is lower than in previous modelling studies and you attribute this to a change in the chemical mechanism. Please add a note here that this point is further explained in section 5.2. Also note that BrCl measurements are discussed in the papers by Lawler et al. (see below). You may want to add that to the discussion in section 4.1.

On page 10 there is a brief discussion about the results by Lawler et al. (2010). First of all, it is either the 2009 or 2011 paper, please check which one you are referring to or add both. Both papers (2009 and 2011) reported high concentrations of HOCl and Cl2 when air was coming from continental Europe, not from Northern Africa as stated here. Please correct your statements. Those papers propose that high HOCl and Cl2 may be caused by aerosol acidification during long range transport and/or slower conversion of HOCl to Cl2 in the aqueous phase. These points are also investigated in Sommariva and von Glasow (2012), which you may want to take into consideration. Is the high HOCl and Cl2 still unexplained in GEOS-Chem if you take into account the findings of these papers?

On page 11: note that the results by Roberts et al. (2008) indicate that the reaction of ClNO2 + Cl- -> Cl2 is relevant only on very acidic aerosol. It may be that aerosol pH during the WINTER campaign was not low enough, which may explain why removing the reaction improves the agreement with the model. Can you please comment? Also, I think the reference is wrong in the bibliography.

MINOR COMMENTS

Page 3: the title of 2.1 should be "GEOS"

Page 7, line 30: add space in "HCl +OH"

Page 12: is the lower Cl* in this work driven by lower BrCl, as mentioned on page 8?

Table 5: there are more observations of ClNO2 available than those listed, especially

in Europe and Asia. Either expand the table or explain why those were chosen. Also, it is not obvious in which order they are listed (year, season, region, concentration?).

Please tidy the references list, there are many mistakes.

---

## Short Comment (SC1) · 2 Dec 2018

Short Comments on Wang X., et al.: The role of chlorine in tropospheric chemistry

1. The authors state on page 5, line 10 that anthropogenic sources of HCl were not included in their base case GEOS-Chem simulation. Although minor in a global sense, it is worth noting that Lee et al. (2018) reported observations of direct halogen (i.e. HCl, as well as $Cl_2$, $ClNO_2$, $Br_2$, $BrNO_2$, and BrCl) emissions from power plants sampled during the WINTER aircraft campaign. This is also important to note in Section 4.2 where model results are directly compared to WINTER chlorine observations.

2. Page 11, line 3, the authors state that WINTER aircraft observations did not extend to the surface layer. As also noted by one Reviewer, there were a series of missed approaches at airfields that could provide further vertical information.

[Figure]

3. On page 10, Table 5 is referred to as a list of 'available' field observations of ClNO2. The GEOS-Chem simulations are then compared to these observations in Table 5 to evaluate the model performance. As noted by both Reviewers, however, there are many additional measurements of ClNO2 that were not included in Table 5. Measurements in addition to those provided by the Reviewers are from: (Edwards et al., 2013; Jeong et al., 2018; Kim et al., 2014; Liu et al., 2017; Osthoff et al., 2008; Phillips et al., 2016; Reyes-Villegas et al., 2018; Tham et al., 2018; Tham et al., 2014; Wang et al., 2018; Wang et al., 2016; Wang et al., 2014; Wild et al., 2016; Yun et al., 2018).

4. The authors include the HOBr + Cl- → BrCl reaction in their mechanism following Abbatt and Waschewsky (1998) and Fickert et al. (1999). They state in the discussion section on page 7 that the Cl source from this reaction is much higher than past simulations. Later in section 5.2, they also include discussion of previous work by Chen at al. (2017) who included a second HOBr + Cl- → Br2 pathway that is dependent on the molar ratio of [Br-]/[Cl-], following Fickert et al. (1999). It is unclear in section 5.2 whether this additional pathway was included in the base case simulation here. If not, the authors should clearly state why it was excluded since the Fickert et al. (1999) laboratory work showed 90% Br2 formation from this reaction at ratios of [Br-]/[Cl-] typically found in ambient sea water. If included, this reaction would help reduce this Cl production pathway relative to previous simulations.

5. The authors have also included the direct reaction of ClNO2 with Cl- to form Cl2 (there is also mention of ClNO2+Br- on page 14, which should be added to Table 2). This reaction is thought to occur via heterogeneous uptake of gas-phase ClNO2 and further reaction with aqueous Cl- (or Br-). As noted by one Reviewer, however, it has been shown by Roberts et al. (2008) that this reaction only occurs at pH < 2 and should be limited here to only highly acidic aerosol. That said, the aerosol during WINTER were highly acidic (pH ∼ -3 to 2) (Guo et al., 2016), which should activate this pathway. Even on these highly acidic aerosol, however, a recent study of WINTER ClNO2 yields by McDuffie et al. (2018a) reported that there was a negative correlation

between particle acidity and CIMS observations of Cl2, which is opposite the trend expected from this reaction. In addition, there was no clear evidence in that study that gas-phase ClNO2 was being lost to heterogeneous processes (reaction with Cl or Br-). Since there is limited field data to support the presence of direct ClNO2 reactions in ambient aerosol, particularly during WINTER, the authors should consider eliminating direct ClNO2 reactions or provide further evidence to support their inclusion in this work.

6. The heterogeneous yield of ClNO2 is only mentioned in reaction R3 in Table 2, where it is defined using a laboratory-based parameterization from Bertram and Thornton (2009). This parameterization is used to predict both N2O5 uptake coefficient and ClNO2 yield. It is concerning that there is no discussion in this manuscript of the large uncertainties associated with these processes or parameterizations. First, this particular parameterization for N2O5 uptake does not consistently reproduce field-derived observations (e.g., Bertram et al., 2009; McDuffie et al., 2018b; Wagner et al., 2013) and has been adjusted in recent GEOS-Chem simulations (Jaeglé et al., 2018; Shah et al., 2018) to better match nitrate observations during WINTER. While N2O5 is not the topic of this manuscript, this process directly impacts the net production of ClNO2, thus impacting the chlorine chemical mechanism and budget. The authors should therefore consider updating the N2O5 uptake parameterization in their simulations or discuss this as a source of uncertainty in their results. Second, this particular parameterization has over-predicted the ClNO2 production yield in every study that has compared its predictions to field-derived results (McDuffie et al., 2018a; Riedel et al., 2013; Ryder et al., 2015; Tham et al., 2018; Thornton et al., 2010; Wagner et al., 2013; Wang, Z. et al., 2017; Wang, X. et al., 2017). In addition, McDuffie et al. (2018a) recently found that the median WINTER ClNO2 production yield was over-predicted by at least 74% by the Bertram and Thornton (2009) parameterization. Since there are no field studies that support this parameterization as written in R3, the authors should adjust this reaction accordingly and discuss its uncertainties.

References

Bertram, T. H., & Thornton, J. A. (2009). Toward a general parameterization of N2O5 reactivity on aqueous particles: the competing effects of particle liquid water, nitrate and chloride. Atmospheric Chemistry and Physics, 9(21), 8351-8363. https://doi.org/10.5194/acp-9-8351-2009

Bertram, T. H., Thornton, J. A., Riedel, T. P., Middlebrook, A. M., Bahreini, R., Bates, T. S., et al. (2009). Direct observations of N2O5 reactivity on ambient aerosol particles. Geophysical Research Letters, 36(19), L19803 https://doi.org/10.1029/2009GL040248

Edwards, P. M., Young, C. J., Aikin, K., deGouw, J., Dubé, W. P., Geiger, F., et al. (2013). Ozone photochemistry in an oil and natural gas extraction region during winter: simulations of a snow-free season in the Uintah Basin, Utah. Atmospheric Chemistry and Physics, 13(17), 8955-8971. https://doi.org/10.5194/acp-13-8955-2013

Guo, H., Sullivan, A. P., Campuzano-Jost, P., Schroder, J. C., Lopez-Hilfiker, F. D., Dibb, J. E., et al. (2016). Fine particle pH and the partitioning of nitric acid during winter in the northeastern United States. Journal of Geophysical Research: Atmospheres, 121(17), 10,355-10,376. https://doi.org/10.1002/2016JD025311

Jaeglé, L., Shah, V., Thornton, J. A., Lopez-Hilfiker, F. D., Lee, B. H., McDuffie, E. E., et al. (2018). Nitrogen Oxides Emissions, Chemistry, Deposition, and Export Over the Northeast United States During the WINTER Aircraft Campaign. Journal of Geophysical Research: Atmospheres, 0(0). https://doi.org/10.1029/2018JD029133

Jeong, D., Seco, R., Gu, D., Lee, Y., Nault, B. A., Knote, C. J., et al. (2018). Integration of Airborne and Ground Observations of Nitryl Chloride in the Seoul Metropolitan Area and the Implications on Regional Oxidation Capacity During KORUS-AQ 2016. Atmos. Chem. Phys. Discuss., 2018, 1-25. https://doi.org/10.5194/acp-2018-1216

Kim, M. J., Farmer, D. K., & Bertram, T. H. (2014). A controlling role for the air−sea interface in the chemical processing of reactive nitrogen in the coastal marine boundary layer. Proceedings of the National Academy of Sciences of the United States of America, 111(11), 3943-3948. https://doi.org/10.1073/pnas.1318694111

Lee, B. H., Lopez-Hilfiker, F. D., Schroder, J. C., Campuzano-Jost, P., Jimenez, J. L., McDuffie, E. E., et al. (2018). Airborne observations of reactive inorganic chlorine and bromine species in the exhaust of coal-fired power plants. Journal of Geophysical Research: Atmospheres, 123. https://doi.org/10.1029/2018JD029284

Liu, X., Qu, H., Huey, L. G., Wang, Y., Sjostedt, S., Zeng, L., et al. (2017). High levels of daytime molecular chlorine and nitryl chloride at a rural site on the North China Plain. Environmental Science & Technology, 51(17), 9588-9595. https://doi.org/10.1021/acs.est.7b03039

McDuffie, E. E., Fibiger, D. L., Dubé, W. P., Lopez Hilfiker, F., Lee, B. H., Jaeglé, L., et al. (2018a). ClNO2 yields from aircraft measurements during the 2015 WINTER campaign and critical evaluation of the current parameterization. Journal of Geophysical Research: Atmospheres, 0(0). https://doi.org/10.1029/2018JD029358

McDuffie, E. E., Fibiger, D. L., Dubé, W. P., Lopez-Hilfiker, F., Lee, B. H., Thornton, J. A., et al. (2018b). Heterogeneous N2O5 uptake duirng winter: Aircraft measurements during the 2015 WINTER campaign and critical evaluation of current parameterizations. Journal of Geophysical Research: Atmospheres, 123(8), 4345-4372. https://doi.org/10.1002/2018JD028336

Osthoff, H. D., Roberts, J. M., Ravishankara, A. R., Williams, E. J., Lerner, B. M., Sommariva, R., et al. (2008). High levels of nitryl chloride in the polluted subtropical marine boundary layer. Nature Geoscience, 1(5), 324-328. https://doi.org/10.1038/ngeo177

Phillips, G. J., Thieser, J., Tang, M., Sobanski, N., Schuster, G., Fachinger, J., et al. (2016). Estimating N2O5 uptake coefficients using ambient measurements of NO3, N2O5, ClNO2 and particle-phase nitrate. Atmospheric Chemistry and Physics, 16(20), 13231-13249. https://doi.org/10.5194/acp-16-13231-2016

Reyes-Villegas, E., Priestley, M., Ting, Y. C., Haslett, S., Bannan, T., Le Breton, M., et al. (2018). Simultaneous aerosol mass spectrometry and chemical ionisation mass spectrometry measurements during a biomass burning event in the UK: insights into nitrate chemistry. Atmospheric Chemistry and Physics, 18(6), 4093-4111. https://doi.org/10.5194/acp-18-4093-2018

Riedel, T. P., Wagner, N. L., Dubé, W. P., Middlebrook, A. M., Young, C. J., Öztürk, F., et al. (2013). Chlorine activation within urban or power plant plumes: Vertically resolved ClNO2 and Cl2 measurements from a tall tower in a polluted continental setting. Journal of Geophysical Research: Atmospheres, 118(15), 8702-8715. https://doi.org/10.1002/jgrd.50637

Roberts, J. M., Osthoff, H. D., Brown, S. S., & Ravishankara, A. R. (2008). N2O5 oxidizes chloride to Cl2 in acidic atmospheric aerosol. Science, 321(5892), 1059. https://doi.org/10.1126/science.1158777

Ryder, O. S., Campbell, N. R., Shaloski, M., Al-Mashat, H., Nathanson, G. M., & Bertram, T. H. (2015). Role of organics in regulating ClNO2 production at the air–sea interface. The Journal of Physical Chemistry A, 119(31), 8519-8526. https://doi.org/10.1021/jp5129673

Shah, V., Jaeglé, L., Thornton, J. A., Lopez-Hilfiker, F. D., Lee, B. H., Schroder, J. C., et al. (2018). Chemical feedbacks weaken the wintertime response of particulate sulfate and nitrate to emissions reductions over the eastern United States. Proceedings of the National Academy of Sciences. https://doi.org/10.1073/pnas.1803295115

Tham, Y. J., Yan, C., Xue, L., Zha, Q., Wang, X., & Wang, T. (2014). Presence of high nitryl chloride in Asian coastal environment and its impact on atmospheric photochemistry. Chinese Science Bulletin, 59(4), 356-359. https://doi.org/10.1007/s11434-013-0063-y

Tham, Y. J., Wang, Z., Li, Q., Wang, W., Wang, X., Lu, K., et al. (2018). Heterogeneous

N2O5 uptake coefficient and production yield of ClNO2 in polluted northern China: Roles of aerosol water content and chemical composition. Atmospheric Chemistry and Physics Discussions, 2018, 1-27. https://doi.org/10.5194/acp-2018-313

Thornton, J. A., Kercher, J. P., Riedel, T. P., Wagner, N. L., Cozic, J., Holloway, J. S., et al. (2010). A large atomic chlorine source inferred from mid-continental reactive nitrogen chemistry. Nature, 464(7286), 271-4. https://doi.org/10.1038/nature08905

Wagner, N. L., Riedel, T. P., Young, C. J., Bahreini, R., Brock, C. A., Dubé, W. P., et al. (2013). N2O5 uptake coefficients and nocturnal NO2 removal rates determined from ambient wintertime measurements. Journal of Geophysical Research: Atmospheres, 118(16), 9331-9350. https://doi.org/10.1002/jgrd.50653

Wang, H., Lu, K., Guo, S., Wu, Z., Shang, D., Tan, Z., et al. (2018). Efficient N2O5 Uptake and NO3 Oxidation in the Outflow of Urban Beijing. Atmospheric Chemistry and Physics Discussions, 2018, 1-27. https://doi.org/10.5194/acp-2018-88

Wang, T., Tham, Y. J., Xue, L., Li, Q., Zha, Q., Wang, Z., et al. (2016). Observations of nitryl chloride and modeling its source and effect on ozone in the planetary boundary layer of southern China. Journal of Geophysical Research: Atmospheres, 121(5), 2476-2489. https://doi.org/10.1002/2015JD024556

Wang, X., Wang, T., Yan, C., Tham, Y. J., Xue, L., Xu, Z., & Zha, Q. (2014). Large daytime signals of N2O5 and NO3 inferred at 62 amu in a TD-CIMS: chemical interference or a real atmospheric phenomenon? Atmospheric Measurement Techniques, 7(1), 1. https://doi.org/10.5194/amt-7-1-2014

Wang, X., Wang, H., Xue, L., Wang, T., Wang, L., Gu, R., et al. (2017). Observations of N2O5 and ClNO2 at a polluted urban surface site in North China: High N2O5 uptake coefficients and low ClNO2 product yields. Atmospheric Environment, 156, 125-134. https://doi.org/10.1016/j.atmosenv.2017.02.035

Wang, Z., Wang, W., Tham, Y. J., Li, Q., Wang, H., Wen, L., et al. (2017). Fast heterogeneous N2O5 uptake and ClNO2 production in power plant and industrial plumes observed in the nocturnal residual layer over the North China Plain. Atmospheric Chemistry and Physics, 17(20), 12361-12378. https://doi.org/10.5194/acp-17-12361-2017

Wild, R. J., Edwards, P. M., Bates, T. S., Cohen, R. C., de Gouw, J. A., Dubé, W. P., et al. (2016). Reactive nitrogen partitioning and its relationship to winter ozone events in Utah. Atmospheric Chemistry and Physics, 16(2), 573-583. https://doi.org/10.5194/acp-16-573-2016

Yun, H., Wang, T., Wang, W., Tham, Y. J., Li, Q., Wang, Z., & Poon, S. C. N. (2018). Nighttime NOx loss and ClNO2 formation in the residual layer of a polluted region: Insights from field measurements and an iterative box model. Science of The Total Environment, 622-623, 727-734. https://doi.org/https://doi.org/10.1016/j.scitotenv.2017.11.352

---

## Author Comment (AC1) · 25 Jan 2019

We thank the reviewers for their time and comments. We have made efforts to improve the manuscript accordingly, please find response for corresponding points below.

Reviewer #1

**This paper presents results describing the application of the GEOS-CHEM model to understanding the role of chlorine chemistry in the troposphere. The paper makes some new contributions and improves upon previous GEOS-CHEM modeling of halogen chemistry, but has so many major flaws and omissions that will require considerable modification. Since these changes may impact some of the findings, it is hard to know how to assess its publishability at this point. I have the following general and specific comments and questions.**

**General Comments:**

**There are number of errors and omissions in the reference list: papers that are noted in the text, but not in list, listed with errors, or just the wrong paper. Here is a list roughly in order of appearance: Liu et al., 2001 – which one, there are two in the reference list? Jaegle et al., 2012 – do you mean Jaegle 2011 here? Wesely 1989 is not in the list. Wang 1998 is not in the list. Jaegle et al., 2010 – do you mean Jaegle 2011 here? Martin et al., 2002 – do you mean 2003? Abbatt and Wachewsky, 1998 is not in the list Bey et al., 2001 is not in the list. Pszenny et al., 1991 – do you mean 1993 here? Roberts et al., 2008 is not in the list. Allen et al., 2007 is not in the list unless you mean Allan et al. 2011 Wang et al., 1994 is not in the list. Roberts et al. 2009 is not the correct reference, it should be this Roberts et al., 2009 (Roberts et al., 2009) Mielke et al., 2013 is not on the list.**

We have corrected the reference list.

**Chloride Sources: The paper ignores soil and wind-blown dust sources of particle chloride. Those were described by Sarwar et al., 2012 (Sarwar et al., 2012) (which by the way is not referenced in this paper) as the major sources of chloride in the Eastern U.S. and therefore will drive the chlorine budget in the middle of continents. This paper needs to give serious consideration to this source, and can use the IMPROVE network chloride data as a means to assess this continental source.**

We now include model comparison to the IMPROVE data (Figure 5) and find that the marine source in the model can largely explain the observed chloride concentrations inland. We now cite Sarwar et al. (2012) for the fugitive dust source and point out that it would be small globally on page 5, line 14-17. We have added "global" to the title to emphasize that our focus is on the global scale.

**Comparisons:**

**The selection of data sets for comparison was certainly not thorough, and obvious opportunities for comparison were apparently not considered. The paper seems to lean heavily on the Graedel and Keene work from 1995, but I believe further work has been done since then. There are comparisons that could be made that would greatly add to the model analysis and there are a number of data sets that were not included in the comparisons that were done.**

**Particle chloride can be compared to IMPROVE data (see for example Sarwar et al., 2012) to see how the model does. This is a long-term data set of considerable geographic extent.**

To respond to the reviewer, we have added comparison to the IMPROVE data with a new Figure (Figure 5).

**There are observations of HCl displacement by HNO3 in the LA Basin that need to be considered for comparison (Gard et al., 1998).**

Observations in the LA Basin are not particularly relevant to us because of our coarse grid resolution ($4^{\circ}$x$5^{\circ}$).

**There are a number of HCl data sets that were not compared with. Here is at least a partial list: Kim et al., INTEX-B (Kim et al., 2008), which includes one of the co-authors. CalNex ground site data for both particle chloride and gas phase HCl can be found at: https://www.esrl.noaa.gov/csd/projects/calnex/.**

We do not include the INTEX-B measurements because Kim et al. (2008) was never published (it has remained in ACPD).

We included CalNex ship data (Crisp et al., 2014) data in Figure 4. These data are better suited for evaluating the model at the relevant resolution than the CalNex ground site.

**There are a considerable number of ClNO2 data sets that were not compared with: Osthoff et al., 2009 – this covers essentially urban/industrial Houston and the near coastal environment. Sommariva et al., 2018 (Sommariva et al., 2018) covers several sites in England, including coastal sites. Le Breton et al., 2018 (Le Breton et al., 2018) Priestley et al., 2018. (Priestley et al., 2018) Tham et al., 2016 (Tham et al., 2013) Wang , Z. et al., (Wang et al., 2017b) Wang X., et al., 2017 (Wang et al., 2017a) Riedel et al, 2013 (Riedel et al., 2013) Mielke et al., 2015 (Mielke et al., 2015) Bannan et al., 2017 (Bannan et al., 2017) Young et al., 2012 (Young et al., 2012)(data set can be found at https://www.esrl.noaa.gov/csd/projects/calnex/) Zhou, et al., 2018 (Zhou, 2018)**

To respond to the reviewer, we have added comparisons with Osthoff et al. (2008), Sommariva et al. (2018), Priestley et al. (2018), Mielke et al. (2015), Bannan et al. (2017), Jeong et al. (2018), Kim et al. (2014), and Tham et al. (2014)  in Table 5. The comparison with Riedel et al. (2013)

was already included. Young et al. (2012) report the same dataset as Mielke et al. (2013) which is already included in the paper. Beyond that, Table 5 is already very long and as we now point out in the paper there is large representation error in trying to evaluate our global model with nighttime urban data (page 10, line 14-18). We also emphasize the global emphasis of our model evaluation at the beginning of Section 4.

**It's true that there is only one lab study of the ClNO2 + Cl- Chemistry (Roberts et al., 2008), but the authors seem to have missed a key point of that paper: that reaction seems to only occur at pH=2 and below. Unless I missed something, this paper seems to allow this reaction after the alkalinity of the aerosol is neutralized. This may account for why the Cl2 is too high compared to the WINTER observations.**

The reviewer did miss something. Table 2 gives $\gamma = 0$ for this reaction when pH > 2. But in any case, the aerosol pH reported by Guo et al. (2016) is smaller than 2 during the WINTER campaign and the same holds in GEOS-Chem. We have added more discussion in Section 4.2, page 12, line 9-15.

**Specific Comments:**

**Page 10, Line 10, and Table 5. There are several data sets for which the model is a factor of 10 low, so it is hard to say the model performed "credibly". Also, most if not all of these data sets are available, so could be compared more in depth, not just maxima.**

We now point out that there is large representation error in comparing our coarse-resolution model with nighttime urban data on page 10, line 14-18. More relevant is our detailed comparison to the ClNO$_2$ WINTER data in Section 4.2.

**Page 11, lines 2&3. I believe the aircraft did missed approaches over airfields that could be used to do near-surface comparisons.**

We deleted the sentence about lack of surface layer observations. These missed-approach data are not of much value to us considering the coarse resolution of the model.

**Page 14, Line 22. Should be 'troposphere'.**

Corrected (typo).

**Reviewer #2**

**This paper by Wang et al. presents an extensive modelling study of chlorine chemistry using GEOS-Chem. The updates to the GEOS-Chem chemical mechanism are much needed. I think the paper is suitable for ACP, after the authors have addressed the following points.**

**MAJOR COMMENTS**

**As a general comment, I think it would be useful to have a more detailed discussion of the difference in the results between the version of GEOS-Chem with the previous mechanism and this new version. As it stands, the reader is mostly referred to previous papers. Figure 9 provides some of this information but it is presented as a sensitivity study, so it is not really a comparison of the old and new mechanisms.**

We have added a supplementary figure S1 and a paragraph to Section 5.3 (page 15, line 25-34) comparing our simulation and GEOS-Chem standard version 11-02d.

**Another general point is that the choice of datasets to compare with the model seems somewhat arbitrary. I understand it is not possible to use all the available datasets, but you have to explain why you selected certain datasets and not others for this work.**

We now emphasize the global emphasis of our model evaluation at the beginning of Section 4 (and have added "global" in the title).We have added comparisons to the IMPROVE Cl⁻ data in new Figure 5 and to additional ClNO$_2$ datasets in Table 5.

**On page 8: you say that the BrCl concentration in this work is lower than in previous modelling studies and you attribute this to a change in the chemical mechanism. Please add a note here that this point is further explained in section 5.2. Also note that BrCl measurements are discussed in the papers by Lawler et al. (see below). You may want to add that to the discussion in section 4.1.**

We have added a note in Section 3 (page 8, line 26) that we would further discuss the lower BrCl in Section 5.2. We have also added the BrCl measurement in Lawler et al. (2009) to the discussion in Section 4.1, page 10, line 3-8.

**On page 10 there is a brief discussion about the results by Lawler et al. (2010). First of all, it is either the 2009 or 2011 paper, please check which one you are referring to or add both. Both papers (2009 and 2011) reported high concentrations of HOCl and Cl2 when air was coming from continental Europe, not from Northern Africa as stated here. Please correct your statements. Those papers propose that high HOCl and Cl2 may be caused by aerosol acidification during long range transport and/or slower conversion of HOCl to Cl2 in the aqueous phase. These points are also investigated in Sommariva and von Glasow (2012),**

**which you may want to take into consideration. Is the high HOCl and Cl2 still unexplained in GEOS-Chem if you take into account the findings of these papers?**

We thank the reviewer for raising this point. We have revised this part on page 10, line 3-13, including:

1) We were referring to Lawler et al. (2011). We now also add the data in Lawler et al. (2009) to the discussion.

2) We have corrected the statement for the sources of high HOCl/$Cl_2$ concentrations.

3) We have cited Sommariva and von Glasow (2012) and added it to the discussion. As Sommariva and von Glasow (2012) pointed out, a lower aerosol pH and/or slower rate for HOCl + $Cl^-$ may explain the high HOCl but would decrease $Cl_2$.

**On page 11: note that the results by Roberts et al. (2008) indicate that the reaction of ClNO2 + Cl- -> Cl2 is relevant only on very acidic aerosol. It may be that aerosol pH during the WINTER campaign was not low enough, which may explain why removing the reaction improves the agreement with the model. Can you please comment? Also, I think the reference is wrong in the bibliography.**

We do consider this pH dependence and it is already listed in Table 2 ($\gamma = 0$ when pH > 2 for R7). In addition, the aerosol pH reported by Guo et al. (2016) are mostly smaller than 2 during WINTER campaign. We have added more discussion for this reaction in Section 4.2, page 12, line 9-15, and corrected the reference.

**MINOR COMMENTS**

**Page 3: the title of 2.1 should be "GEOS"**

Changed (typo).

**Page 7, line 30: add space in "HCl +OH"**

Added.

**Page 12: is the lower Cl* in this work driven by lower BrCl, as mentioned on page 8?**

The lower Cl* is driven by both the lower rate of $Cl_y$ generation from acid displacement (as discussed in Section 3) and lower BrCl.

**Table 5: there are more observations of ClNO2 available than those listed, especially in Europe and Asia. Either expand the table or explain why those were chosen. Also, it is not obvious in which order they are listed (year, season, region, concentration?).**

We have added more datasets in our comparison of ClNO$_2$ in Table 5. We also add a statement at the beginning of Section 4 to emphasize that we only include data those not heavily affected by local anthropogenic sources. The order is based on latitude (from north to south), this is now explained in the Table footnotes.

**Please tidy the references list, there are many mistakes.**

The references list has been corrected.

**Short Comments by McDuffie**

**1. The authors state on page 5, line 10 that anthropogenic sources of HCl were not included in their base case GEOS-Chem simulation. Although minor in a global sense, it is worth noting that Lee et al. (2018) reported observations of direct halogen (i.e. HCl, as well as Cl2, ClNO2, Br2, BrNO2, and BrCl) emissions from power plants sampled during the WINTER aircraft campaign. This is also important to note in Section 4.2 where model results are directly compared to WINTER chlorine observations.**

We now include the power plant source of HCl following Lee et al. (2018) in a sensitivity simulation. This is discussed in Section 4.2, page 11, line 10-15.

**2. Page 11, line 3, the authors state that WINTER aircraft observations did not extend to the surface layer. As also noted by one Reviewer, there were a series of missed approaches at airfields that could provide further vertical information.**

We deleted the sentence about lack of surface layer observations. These missed-approach data are not of much value to us considering the coarse resolution of the model.

**3. On page 10, Table 5 is referred to as a list of 'available' field observations of ClNO2. The GEOS-Chem simulations are then compared to these observations in Table 5 to evaluate the model performance. As noted by both Reviewers, however, there are many additional measurements of ClNO2 that were not included in Table 5. Measurements in addition to those provided by the Reviewers are from: (Edwards et al., 2013; Jeong et al., 2018; Kim et al., 2014; Liu et al., 2017; Osthoff et al., 2008; Phillips et al., 2016; Reyes-Villegas et al., 2018; Tham et al., 2018; Tham et al., 2014; Wang et al., 2018; Wang et al., 2016; Wang et al., 2014; Wild et al., 2016; Yun et al., 2018).**

Thank you for pointing out these measurement data. We have added the comparisons with Osthoff et al. (2008), Sommariva et al. (2018), Priestley et al. (2018), Mielke et al. (2015), Bannan et al. (2017), Jeong et al. (2018), Kim et al. (2014), and Tham et al. (2014) in Table 5. Phillips et al. (2016) used the exactly same dataset as Phillips et al. (2012), which was already included in the paper. Wang et al. (2016) was also already included in the paper. Yun et al. (2018) used the exactly same dataset as Wang et al. (2016). There is no measurement of $ClNO_2$ concentration presented in Wang et al. (2014).

As pointed out in response to Reviewer 1 and in the revised text on page 10, line 14-18, comparison to nighttime urban $ClNO_2$ data is of limited interest for model evaluation because of the resolution mismatch. We also emphasize the global emphasis of our model evaluation at the beginning of Section 4.

**4. The authors include the HOBr + Cl- ! BrCl reaction in their mechanism following Abbatt and Waschewsky (1998) and Fickert et al. (1999). They state in the discussion section on page 7 that the Cl source from this reaction is much higher than past simulations. Later in section 5.2, they also include discussion of previous work by Chen at al. (2017) who included a second HOBr + Cl- ! Br2 pathway that is dependent on the molar ratio of [Br-]/[Cl-], following Fickert et al. (1999). It is unclear in section 5.2 whether this additional pathway was included in the base case simulation here. If not, the authors should clearly state why it was excluded since the Fickert et al. (1999) laboratory work showed 90% Br2 formation from this reaction at ratios of [Br-]/[Cl-] typically found in ambient sea water. If included, this reaction would help reduce this Cl production pathway relative to previous simulations.**

We have clarified on page 14, line 8 that we include this mechanism in our model simulation. We have also mentioned this mechanism follows Fickert et al. (1999) and would be further described in section 5.2 in the footnote of Table 2.

**5. The authors have also included the direct reaction of ClNO2 with Cl- to form Cl2 (there is also mention of ClNO2+Br- on page 14, which should be added to Table 2. This reaction is thought to occur via heterogeneous uptake of gas-phase ClNO2 and further reaction with aqueous Cl- (or Br-). As noted by one Reviewer, however, it has been shown by Roberts et al. (2008) that this reaction only occurs at pH < 2 and should be limited here to only highly acidic aerosol. That said, the aerosol during WINTER were highly acidic (pH = -3 to 2) (Guo et al., 2016), which should activate this pathway. Even on these highly acidic aerosol, however, a recent study of WINTER ClNO2 yields by McDuffie et al. (2018a) reported that there was a negative correlation between particle acidity and CIMS observations of Cl2, which is opposite the trend expected from this reaction. In addition, there was no clear evidence in that study that gas-phase ClNO2 was being lost to heterogeneous processes (reaction with Cl or Br-). Since there is limited field data to support the presence of direct ClNO2 reactions in ambient aerosol, particularly during WINTER, the authors should consider eliminating direct ClNO2 reactions or provide further evidence to support their inclusion in this work.**

Thank you for raising these points. We have added $ClNO_2 + Br^-$ as R8 in Table 2.

We do consider the pH dependence of $ClNO_2 + Cl^-$ and it is already listed in Table 2 ($\gamma = 0$ when pH > 2 for R7). We also add more discussion in this section on page 12, line 9-15 citing McDuffie et al. (2018).

**6. The heterogeneous yield of ClNO2 is only mentioned in reaction R3 in Table 2, where it is defined using a laboratory-based parameterization from Bertram and Thornton (2009). This parameterization is used to predict both N2O5 uptake coefficient and ClNO2 yield. It is concerning that there is no discussion in this manuscript of the large uncertainties associated with these processes or parameterizations. First, this particular parameterization for N2O5 uptake does not consistently reproduce field-derived observations (e.g., Bertram et al., 2009;**

**McDuffie et al., 2018b; Wagner et al., 2013) and has been adjusted in recent GEOS-Chem simulations (Jaeglé et al., 2018; Shah et al., 2018) to better match nitrate observations during WINTER. While N2O5 is not the topic of this manuscript, this process directly impacts the net production of ClNO2, thus impacting the chlorine chemical mechanism and budget. The authors should therefore consider updating the N2O5 uptake parameterization in their simulations or discuss this as a source of uncertainty in their results. Second, this particular parameterization has over-predicted the ClNO2 production yield in every study that has compared its predictions to field-derived results (McDuffie et al., 2018a; Riedel et al., 2013; Ryder et al., 2015; Tham et al., 2018; Thornton et al., 2010; Wagner et al., 2013; Wang, Z. et al., 2017; Wang, X. et al., 2017). In addition, McDuffie et al. (2018a) recently found that the median WINTER ClNO2 production yield was over-predicted by at least 74% by the Bertram and Thornton (2009) parameterization. Since there are no field studies that support this parameterization as written in R3, the authors should adjust this reaction accordingly and discuss its uncertainties.**

Jaeglé et al. (2018) and Shah et al. (2018) actually used the Bertram and Thornton (2009) scheme with no adjustment. Both papers updated the $N_2O_5$ uptake from old GEOS-Chem versions to the exactly same scheme as in our paper and found better match to the nitrate observations during WINTER. A major reason for the difference with McDuffie et al. is likely that GEOS-Chem treats reactive uptake by aerosols assuming an external mixture.

We have added more discussion for the $N_2O_5$ uptake parametrization and give our tentative explanation for the difference with McDuffie et al. on page 11 line 29 – page 12 line 4.

---

## Referee Report (RR1)

Response to Author Revisions

The authors have thoughtfully modified the manuscript to address many of the comments received during the review. There remain three sections that should be addressed prior to publication.

Page 11, Line 29

The authors have chosen to retain the $\phi(ClNO_2)$ parameterization based on previous laboratory results. The authors state that agreement of resulting simulated $ClNO_2$ with WINTER observations is in contrast with previous results (McDuffie et al., 2018a; Riedel et al., 2013; Wagner et al., 2013), that have shown the Bertram and Thornton (2009) parameterization over-predicts $ClNO_2$ production. Two minor points, 1) the cited references report that the *yield of $ClNO_2$* ($\phi(ClNO_2)$) is over-predicted by the parameterization, not necessarily the absolute amount of $ClNO_2$, 2) only McDuffie et al., 2018a ($ClNO_2$ yields) should be cited here as McDuffie et al., 2018b only reports $\gamma(N_2O_5)$ results from WINTER.

My bigger concern is that the authors have not sufficiently discussed the extent of the disagreement that exists in the literature between field and laboratory-derived $\phi(ClNO_2)$ results. I want to reiterate that while all laboratory-based studies have reported similar parametrizations of $\phi(ClNO_2)$ (Behnke et al., 1997; Bertram & Thornton, 2009; Roberts et al., 2009; Ryder et al., 2015), *every single* field study has shown that this $\phi(ClNO_2)$ parameterization over-predicts $\phi(ClNO_2)$ on ambient aerosol (see references in my initial review). There is currently no consensus on the source of this disagreement. In this analysis, it remains concerning that the authors have chosen to implement a parameterization that has been repeatedly shown to disagree with field results, especially without providing sufficient recognition of this disagreement or sufficient motivation for its inclusion.

Page 11, Line 33

The added discussion of $\gamma(N_2O_5)$ and $\phi(ClNO_2)$ results from (McDuffie et al., 2018a; McDuffie et al., 2018b) is useful here as these values were derived for the same campaign that is used to evaluate the model performance here. First, it would be better, in my opinion, to recognize the uncertainty in the $\gamma(N_2O_5)$ parameterization (as the authors have done), but then cite the $\gamma(N_2O_5)$ agreement between GEOS-Chem and the McDuffie et al. (2018b) box model that was previously presented in Jaeglé et al. (2018). As the authors have noted in their response, the $\gamma(N_2O_5)$ parameterization is the same here in Jaeglé et al. (2018) (though the particle chloride concentrations were calculated differently and Jaeglé et al. (2018)/Shah et al. (2018) also updated the organic aerosol uptake coefficient), so the reported agreement in $\gamma(N_2O_5)$ could help to validate the $\gamma(N_2O_5)$ calculation here.

My second comment is that the discussion of mixing state and chloride distributions across different aerosol types does not explain the overestimation of $\gamma(N_2O_5)$ or $\phi(ClNO_2)$ by Bertram and Thornton (2009) parameterization that was reported by (McDuffie et al., 2018a; McDuffie et al., 2018b), nor does it help validate their use here. For instance, since the $\gamma(N_2O_5)$ and $\phi(ClNO_2)$ parameterizations do not explicitly account of organic aerosol concentrations, $\gamma(N_2O_5)$ and $\phi(ClNO_2)$ would only be lower here relative to the McDuffie results if there were

additional organic-associated chloride that was not being accounted for in GEOS-Chem by the assumption that chloride is only present in the SSA and SNA aerosol types. In addition, the authors may be further complicating this discussion by introducing the concept of mixing state because Bertram and Thornton (2009) explicitly state that their parameterization was 'designed to address an internally mixed particle population' (granted, SNA particles are assumed internally mixed in GEOS-Chem). Rather than discussing mixing state assumptions, I suggest simply citing and Jaeglé et al. (2018) to validate the $\gamma(N_2O_5)$ parameterization as described above.

Page 12, line 9 –
The discussion of the $ClNO_2 + Cl^-$ → $Cl_2$ reaction has greatly improved with the addition of the sensitivity test where R7 is removed. May I also suggest citing Fickert et al. (1998), Schweitzer et al. (1998), and Frenzel et al. (1998) who report $ClNO_2$ uptake coefficients for similar reactions with $Br^-$ and $I^-$ that are ~ 2 orders of magnitude lower than coefficients presented in Roberts et al. (2008). This could help to provide additional context for the magnitude of direct $ClNO_2$ uptake/reactions.

Minor Comments:
Page 7, line 7/8: Add reference to Roberts et al. (2008).
I would also suggest changing this sentence to: "The heterogeneous uptake of HOBr, HOCl, and $ClNO_2$, and further aqueous-phase reaction with $Cl^-$ has been shown to be pH-dependent, with a higher efficiency in acidic solutions.

Page 7, line 8/9 -
Change to: "They are considered… , and the reaction of $ClNO_2 + Cl^-$ further requires pH < 2, following laboratory results presented in Roberts et al. (2008)."

Page 9, line 25 –
remove the extra period at the end of the sentence.

Page 10, line 25 –
Change to: "…by an Iodide, Time of Flight, Chemical Ionization Mass Spectrometer ($I^-$TOF-CIMS).

Line 14 –
Suggest adding: "… is opposite of the trend expected from (R7), though there may be no trend on sufficiently acidic (i.e., pH < 2) aerosol.

**References**

Behnke, W., George, C., Scheer, V., & Zetzsch, C. (1997). Production and decay of $ClNO_2$ from the reaction of gaseous $N_2O_5$ with NaCl solution: Bulk and aerosol experiments. *Journal of Geophysical Research: Atmospheres, 102*(D3), 3795-3804. https://doi.org/10.1029/96JD03057

Bertram, T. H., & Thornton, J. A. (2009). Toward a general parameterization of $N_2O_5$ reactivity on aqueous particles: the competing effects of particle liquid water, nitrate and chloride. *Atmospheric Chemistry and Physics, 9*(21), 8351-8363. https://doi.org/10.5194/acp-9-8351-2009

Fickert, S., Helleis, F., Adams, J. W., Moortgat, G. K., & Crowley, J. N. (1998). Reactive uptake of $ClNO_2$ on aqueous bromide solutions. *Journal of Physical Chemistry A, 102*(52), 10689-10696. https://doi.org/10.1021/jp983004n

Frenzel, A., Scheer, V., Sikorski, R., George, C., Behnke, W., & Zetzsch, C. (1998). Heterogeneous Interconversion Reactions of $BrNO_2$, $ClNO_2$, $Br_2$, and $Cl_2$. *The Journal of Physical Chemistry A, 102*(8), 1329-1337. https://doi.org/10.1021/jp973044b

Jaeglé, L., Shah, V., Thornton, J. A., Lopez-Hilfiker, F. D., Lee, B. H., McDuffie, E. E., et al. (2018). Nitrogen Oxides Emissions, Chemistry, Deposition, and Export Over the Northeast United States During the WINTER Aircraft Campaign. *Journal of Geophysical Research: Atmospheres, 123*(21), 12,368-12,393. https://doi.org/10.1029/2018JD029133

McDuffie, E. E., Fibiger, D. L., Dubé, W. P., Lopez Hilfiker, F., Lee, B. H., Jaeglé, L., et al. (2018a). $ClNO_2$ Yields From Aircraft Measurements During the 2015 WINTER Campaign and Critical Evaluation of the Current Parameterization. *Journal of Geophysical Research: Atmospheres, 123*(22), 12,994-13,015. https://doi.org/10.1029/2018JD029358

McDuffie, E. E., Fibiger, D. L., Dubé, W. P., Lopez-Hilfiker, F., Lee, B. H., Thornton, J. A., et al. (2018b). Heterogeneous $N_2O_5$ uptake duirng winter: Aircraft measurements during the 2015 WINTER campaign and critical evaluation of current parameterizations. *Journal of Geophysical Research: Atmospheres, 123*(8), 4345-4372. https://doi.org/10.1002/2018JD028336

Riedel, T. P., Wagner, N. L., Dubé, W. P., Middlebrook, A. M., Young, C. J., Öztürk, F., et al. (2013). Chlorine activation within urban or power plant plumes: Vertically resolved $ClNO_2$ and $Cl_2$ measurements from a tall tower in a polluted continental setting. *Journal of Geophysical Research: Atmospheres, 118*(15), 8702-8715. https://doi.org/10.1002/jgrd.50637

Roberts, J. M., Osthoff, H. D., Brown, S. S., & Ravishankara, A. R. (2008). $N_2O_5$ oxidizes chloride to $Cl_2$ in acidic atmospheric aerosol. *Science, 321*(5892), 1059. https://doi.org/10.1126/science.1158777

Roberts, J. M., Osthoff, H. D., Brown, S. S., Ravishankara, A. R., Coffman, D., Quinn, P., & Bates, T. (2009). Laboratory studies of products of $N_2O_5$ uptake on $Cl^-$ containing substrates. *Geophysical Research Letters, 36*(20), L20808. https://doi.org/10.1029/2009GL040448

Ryder, O. S., Campbell, N. R., Shaloski, M., Al-Mashat, H., Nathanson, G. M., & Bertram, T. H. (2015). Role of organics in regulating $ClNO_2$ production at the air–sea interface. *The Journal of Physical Chemistry A, 119*(31), 8519-8526. https://doi.org/10.1021/jp5129673

Schweitzer, F., Mirabel, P., & George, C. (1998). Multiphase Chemistry of $N_2O_5$, $ClNO_2$, and $BrNO_2$. *The Journal of Physical Chemistry A, 102*(22), 3942-3952. https://doi.org/10.1021/jp980748s

Shah, V., Jaeglé, L., Thornton, J. A., Lopez-Hilfiker, F. D., Lee, B. H., Schroder, J. C., et al. (2018). Chemical feedbacks weaken the wintertime response of particulate sulfate and nitrate to emissions reductions over the eastern United States. *Proceedings of the National Academy of Sciences*. https://doi.org/10.1073/pnas.1803295115

Wagner, N. L., Riedel, T. P., Young, C. J., Bahreini, R., Brock, C. A., Dubé, W. P., et al. (2013). $N_2O_5$ uptake coefficients and nocturnal $NO_2$ removal rates determined from ambient wintertime measurements. *Journal of Geophysical Research: Atmospheres, 118*(16), 9331-9350. https://doi.org/10.1002/jgrd.50653

---

## Author Response (AR2)

We thank both reviewers for their time and comments. We have made efforts to improve the manuscript accordingly, please find response for corresponding points below.

**Reviewer #1**

**The authors answered most of the previous comments/issues except for the following:**

**The authors point out the fact that Kim et al., 2008 was not carried through to publication as the reason for not doing a comparison. Aren't the data still available on the NASA data site? Are the authors (one of whom was the senior author on the Kim paper) saying that they don't believe the HCl measurements? They should state that or do the comparison.**

We now include model comparison to INTEX-B data in Figure 8 and in the fourth paragraph on page 11.

**The authors are mistaken that Young et al. 2012 only repeated the data of Mielke et al., 2013. The Young et al., paper, and it turns out Wagner et al., 2012, also have vertical profiles from the NOAA WP3 measurements that were part of CalNex 2010. Again, the whole data set is available: https://www.esrl.noaa.gov/csd/projects/calnex/.**

There is no chlorine measurement available to public from the NOAA WP3 measurements during CalNex 2010 (https://esrl.noaa.gov/csd/groups/csd7/measurements/2010calnex/P3/datainfo.html). We do not include this dataset.

**Reviewer #2**

**The authors have thoughtfully modified the manuscript to address many of the comments received during the review. There remain three sections that should be addressed prior to publication.**

**Page 11, Line 29**

**The authors have chosen to retain the $\Phi(ClNO2)$ parameterization based on previous laboratory results. The authors state that agreement of resulting simulated ClNO2 with WINTER observations is in contrast with previous results (McDuffie et al., 2018a; Riedel et al., 2013; Wagner et al., 2013), that have shown the Bertram and Thornton (2009) parameterization over-predicts ClNO2 production. Two minor points, 1) the cited references report that the *yield of ClNO2* ($\Phi(ClNO2)$) is over-predicted by the parameterization, not necessarily the absolute amount of ClNO2, 2) only McDuffie et al., 2018a (ClNO2 yields) should be cited here as McDuffie et al., 2018b only reports $\gamma(N2O5)$ results from WINTER.**

We now clarify these points on page 11 line 35 – page 12 line 2.

**My bigger concern is that the authors have not sufficiently discussed the extent of the disagreement that exists in the literature between field and laboratory-derived Φ (ClNO2) results. I want to reiterate that while all laboratory-based studies have reported similar parametrizations of Φ (ClNO2) (Behnke et al., 1997; Bertram & Thornton, 2009; Roberts et al., 2009; Ryder et al., 2015), *every single* field study has shown that this Φ (ClNO2) parameterization over-predicts Φ (ClNO2) on ambient aerosol (see references in my initial review). There is currently no consensus on the source of this disagreement. In this analysis, it remains concerning that the authors have chosen to implement a parameterization that has been repeatedly shown to disagree with field results, especially without providing sufficient recognition of this disagreement or sufficient motivation for its inclusion.**

As already stated in our last response, although our parameterization of N2O5 uptake on SNA is based on Bertram and Thornton, (2009), our simulation of N2O5 uptake and ClNO2 yield also combines the uptake on other aerosols. By assuming externally mixing of aerosol types when doing N2O5 uptake, the overall N2O5 uptake rate and ClNO2 yield on all aerosols are different from Bertram and Thornton, (2009). The mean ClNO2 yield below 1 km in our model is 0.20 during WINTER campaign. This agrees with the field results of McDuffie et al., (2018a) (0.218). We now edit the text in the first paragraph on page 12 and add a note in the footnote in Table 2 to make it clearer.

Page 11, Line 33

**The added discussion of γ(N2O5) and Φ (ClNO2) results from (McDuffie et al., 2018a; McDuffie et al., 2018b) is useful here as these values were derived for the same campaign that is used to evaluate the model performance here. First, it would be better, in my opinion, to recognize the uncertainty in the γ(N2O5) parameterization (as the authors have done), but then cite the γ(N2O5) agreement between GEOS-Chem and the McDuffie et al. (2018b) box model that was previously presented in Jaeglé et al. (2018). As the authors have noted in their response, the γ(N2O5) parameterization is the same here in Jaeglé et al. (2018) (though the particle chloride concentrations were calculated differently and Jaeglé et al. (2018)/Shah et al. (2018) also updated the organic aerosol uptake coefficient), so the reported agreement in γ(N2O5) could help to validate the γ(N2O5) calculation here.**

We now cite Jaeglé et al. (2018) and edit the text in the first paragraph on page 12 to make our comparison clearer.

**My second comment is that the discussion of mixing state and chloride distributions across different aerosol types does not explain the overestimation of γ(N2O5) or Φ(ClNO2) by Bertram and Thornton (2009) parameterization that was reported by (McDuffie et al., 2018a; McDuffie et al., 2018b), nor does it help validate their use here. For instance, since the γ(N2O5) and Φ(ClNO2) parameterizations do not explicitly account of organic aerosol concentrations, γ(N2O5) and Φ(ClNO2) would only be lower here relative to the McDuffie**

**results if there were additional organic-associated chloride that was not being accounted for in GEOS-Chem by the assumption that chloride is only present in the SSA and SNA aerosol types. In addition, the authors may be further complicating this discussion by introducing the concept of mixing state because Bertram and Thornton (2009) explicitly state that their parameterization was 'designed to address an internally mixed particle population' (granted, SNA particles are assumed internally mixed in GEOS-Chem). Rather than discussing mixing state assumptions, I suggest simply citing and Jaeglé et al. (2018) to validate the γ(N2O5) parameterization as described above.**

The discussion of mixing state is not used for explaining the overestimation of N2O5 or ClNO2 by Bertram and Thornton (2009) parameterization. We include the discussion of mixing state to state that our model parametrization of N2O5 uptake on all aerosols is different from Bertram and Thornton (2009) parameterization. We agree with the reviewer regarding the overestimation of Bertram and Thornton (2009) parameterization. We have edited the text in the first paragraph on page 12 to make this clearer.

**Page 12, line 9 –**

**The discussion of the ClNO2 + Cl- → Cl2 reaction has greatly improved with the addition of the sensitivity test where R7 is removed. May I also suggest citing Fickert et al. (1998), Schweitzer et al. (1998), and Frenzel et al. (1998) who report ClNO2 uptake coefficients for similar reactions with Br- and I- that are ~ 2 orders of magnitude lower than coefficients presented in Roberts et al. (2008). This could help to provide additional context for the magnitude of direct ClNO2 uptake/reactions.**

We add a sentence and cite these references on page 12, line 22-24.

**Minor Comments:**

**Page 7, line 7/8: Add reference to Roberts et al. (2008).**

**I would also suggest changing this sentence to: "The heterogeneous uptake of HOBr, HOCl, and ClNO2, and further aqueous-phase reaction with Cl- has been shown to be pH-dependent, with a higher efficiency in acidic solutions.**

We have referenced Roberts et al. (2008) and changed this sentence following the reviewer's suggestion.

**Page 7, line 8/9 -**

**Change to: "They are considered… , and the reaction of ClNO2+ Cl- further requires pH < 2, following laboratory results presented in Roberts et al. (2008)."**

Changed.

**Page 9, line 25 –**

**remove the extra period at the end of the sentence.**

Removed.

**Page 10, line 25 –**

**Change to: "…by an Iodide, Time of Flight, Chemical Ionization Mass Spectrometer (I-TOF-CIMS).**

Changed.

**Line 14 –**

[revised manuscript text omitted]

Section 4.1.

Mixing ratios of $ClNO_2$ observed in WINTER are above the detection limit only in the lowest km of atmosphere at night, and are much higher over the ocean than over land.  This is well simulated by the model (Figures 6 and 7), and reflects the nighttime source from the $N_2O_5 + Cl^-$ heterogeneous reaction from the combined with the fast loss by photolysis in daytime. Previous studies have suggested that the Bertram and Thornton (2009) representation of the

ClNO$_2$ production yield from the N$_2$O$_5$ + Cl$^-$ heterogeneous reaction in Table 2 is too high (Riedel et al., 2013; Wagner et al., 2013; McDuffie et al., 2018a). By using a box model applied to the WINTER observations, McDuffie et al. (2018a;b) found that both the N$_2$O$_5$ uptake rate and the ClNO$_2$ production yield were overestimated by the Bertram and Thornton (2009) parameterization. One important difference with these previous studies is the assumption of aerosol mixing state. GEOS-Chem assumes that Cl$^-$ is present only in SNA and SSA when doing the calculation of N$_2$O$_5$ reactive uptake rates, assuming an external mixture of aerosol types (Martin et al., 2003; Evans and Jacob, 2005; Jaeglé et al., 2018). This decreases both the N$_2$O$_5$ uptake rate and the ClNO$_2$ yield as compared to using the parameterization of Bertram and Thornton (2009) on internally mixed aerosols. Jaeglé et al. (2018) reported that using this external mixing assumption leads to good agreement between modeled and observed N$_2$O$_5$ during WINTER campaign. The mean ClNO$_2$ yield ($\varphi$ in R3) below 1 km in our model is 0.20 during the WINTER campaign, very close to that calculated from observation (0.218) by McDuffie et al., (2018a).

Nighttime Cl$_2$ mixing ratios in WINTER are greatly overestimated by the model. Under polluted wintertime conditions such as in WINTER the ClNO$_2$ + Cl$^-$ reaction greatly enhances Cl$_2$ production in the model:

$$ClNO_2 + Cl^- \rightarrow NO_2^- + Cl_2 \qquad\qquad (R7)$$

The reactive uptake coefficient for (R7) in Table 2 is based on a single laboratory study (Roberts et al., 2008). It requires an aerosol pH < 2 and this condition is generally met for our model simulation of the WINTER environment, consistent with the observation-based analysis of aerosol pH by Guo et al. (2016) for the eastern US in winter. A sensitivity simulation without (R7) is shown as dashed red lines in Figure 7 and can reproduce the low Cl$_2$ mixing ratios observed over the ocean at night. The analysis of WINTER data by McDuffie et al. (2018b) finds that the correlation between particle acidity and Cl$_2$ observations is opposite of the trend expected from (R7), though there may be no trend on sufficiently acidic (i.e. pH < 2) aerosol. Further study of that reaction is needed. Similar reactions such as ClNO$_2$ + Br$^-$ (R8) and ClNO$_2$ + I$^-$ are also very uncertain as their reaction rate coefficients are very different in different studies (Fickert et al., 1998; Frenzel et al., 1998; Schweitzer et al., 1998).

[revised manuscript text omitted]
bromide solutions. Journal of Physical Chemistry A, 102(52), 10689-10696. https://doi.org/10.1021/jp983004n, 1998.

Fickert, S., Adams, J. W., and Crowley, J. N.: Activation of Br2 and BrCl via uptake of HOBr onto aqueous salt
solutions, Journal of Geophysical Research: Atmospheres, 104, 23719-23727, doi:10.1029/1999JD900359, 1999.

Finlayson-Pitts, B. J.: The Tropospheric Chemistry of Sea Salt: A Molecular-Level View of the Chemistry of NaCl
and NaBr, Chemical Reviews, 103, 4801-4822, 10.1021/cr020653t, 2003.

Fountoukis, C., and Nenes, A.: ISORROPIA II: a computationally efficient thermodynamic equilibrium model for
$K^+–Ca^{2+}–Mg^{2+}–NH_4^+–Na^+–SO_4^{2-}–NO_3^-–Cl^-–H_2O$ aerosols,, Atmos. Chem. Phys., 7, 4639-4659, 10.5194/acp-7-
4639-2007, 2007.

Frenzel, A., Scheer, V., Sikorski, R., George, C., Behnke, W., & Zetzsch, C.: Heterogeneous Interconversion
Reactions of BrNO2, ClNO2, Br2, and Cl2. The Journal of Physical Chemistry A, 102(8), 1329-1337.
https://doi.org/10.1021/jp973044b, 1998.

[revised manuscript text omitted]

Jaeglé, L., Shah, V., Thornton, J. A., Lopez-Hilfiker, F. D., Lee, B. H., McDuffie, E. E., Fibiger, D., Brown, S. S., Veres, P., Sparks, T. L., Ebben, C. J., Wooldridge, P. J., Kenagy, H. S., Cohen, R. C., Weinheimer, A. J., Campos, T.

L., Montzka, D. D., Digangi, J. P., Wolfe, G. M., Hanisco, T., Schroder, J. C., Campuzano-Jost, P., Day, D. A., Jimenez, J. L., Sullivan, A. P., Guo, H., and Weber, R. J.: Nitrogen Oxides Emissions, Chemistry, Deposition, and Export Over the Northeast United States During the WINTER Aircraft Campaign, Journal of Geophysical Research: Atmospheres, 123, 12,368-312,393, doi:10.1029/2018JD029133, 2018.

Jeong, D., Seco, R., Gu, D., Lee, Y., Nault, B. A., Knote, C. J., McGee, T., Sullivan, J. T., Jimenez, J. L., Campuzano-Jost, P., Blake, D. R., Sanchez, D., Guenther, A. B., Tanner, D., Huey, L. G., Long, R., Anderson, B. E., Hall, S. R., Ullmann, K., Shin, H. J., Herndon, S. C., Lee, Y., Kim, D., Ahn, J., and Kim, S.: Integration of Airborne and Ground Observations of Nitryl Chloride in the Seoul Metropolitan Area and the Implications on Regional Oxidation Capacity During KORUS-AQ 2016, Atmos. Chem. Phys. Discuss., 2018, 1-25, 10.5194/acp-2018-1216, 2018.

Kasibhatla, P., Sherwen, T., Evans, M. J., Carpenter, L. J., Reed, C., Alexander, B., Chen, Q., Sulprizio, M. P., Lee, J. D., Read, K. A., Bloss, W., Crilley, L. R., Keene, W. C., Pszenny, A. A. P., and Hodzic, A.: Global impact of nitrate photolysis in sea-salt aerosol on NOx, OH, and O3 in the marine boundary layer, Atmos. Chem. Phys., 18, 11185-11203, 10.5194/acp-18-11185-2018, 2018.

Keene, W. C., Pszenny, A. A. P., Jacob, D. J., Duce, R. A., Galloway, J. N., Schultz-Tokos, J. J., Sievering, H., and Boatman, J. F.: The geochemical cycling of reactive chlorine through the marine troposphere, Global Biogeochemical Cycles, 4, 407-430, doi:10.1029/GB004i004p00407, 1990.

Keene, W. C., Stutz, J., Pszenny, A. A. P., Maben, J. R., Fischer, E. V., Smith, A. M., von Glasow, R., Pechtl, S., Sive, B. C., and Varner, R. K.: Inorganic chlorine and bromine in coastal New England air during summer, Journal of Geophysical Research: Atmospheres, 112, 10.1029/2006jd007689, 2007.

Keene, W. C., Long, M. S., Pszenny, A. A. P., Sander, R., Maben, J. R., Wall, A. J., O'Halloran, T. L., Kerkweg, A., Fischer, E. V., and Schrems, O.: Latitudinal variation in the multiphase chemical processing of inorganic halogens and related species over the eastern North and South Atlantic Oceans, Atmos. Chem. Phys., 9, 7361-7385, 10.5194/acp-9-7361-2009, 2009.

Kelly, J. T., Bhave, P. V., Nolte, C. G., Shankar, U., and Foley, K. M.: Simulating emission and chemical evolution of coarse sea-salt particles in the Community Multiscale Air Quality (CMAQ) model, Geosci. Model Dev., 3, 257-273, 10.5194/gmd-3-257-2010, 2010.

Kercher, J. P., Riedel, T. P., and Thornton, J. A.: Chlorine activation by N2O5: simultaneous, in situ detection of ClNO2 and N2O5 by chemical ionization mass spectrometry, Atmos. Meas. Tech., 2, 193-204, 10.5194/amt-2-193-2009, 2009.

Kim, S., Huey, L. G., Stickel, R. E., Pierce, R. B., Chen, G., Avery, M. A., Dibb, J. E., Diskin, G. S., Sachse, G. W., McNaughton, C. S., Clarke, A. D., Anderson, B. E., and Blake, D. R.: Airborne measurements of HCl from the marine boundary layer to the lower stratosphere over the North Pacific Ocean during INTEX-B, Atmos. Chem. Phys. Discuss., 8, 3563-3595, https://doi.org/10.5194/acpd-8-3563-2008, 2008.

[revised manuscript text omitted]

**Chlorine driven changes in BrO, NO$_x$, OH, and ozone**

[Figure]

**Figure 9: Effects of tropospheric chlorine chemistry on BrO, NO$_x$, OH, and ozone concentrations. The top panels show the annual mean surface concentrations of BrO, NO$_x$, OH, and ozone simulated in our standard model including tropospheric chlorine chemistry. The lower panels show the changes in annual mean mixing ratios/concentrations due to tropospheric chlorine chemistry, as determined by difference with a sensitivity simulation including no Cl$_y$ production and cycling. The middle panels show the changes in surface air concentrations and the bottom panels show the changes in zonal mean mixing concentrations as a function of latitude and altitude. Black dashed lines indicate the tropopause. Numbers in bottom panels show the global tropospheric mean differences.**